# Convergence Guarantees for Gradient-Based Training of Neural PDE Solvers: From Linear to Nonlinear PDEs

## Abstract

We present a unified convergence theory for gradient-based training of neural network methods for partial differential equations (PDEs), covering both physics-informed neural networks (PINNs) and the Deep Ritz method. For linear PDEs, we extend the neural tangent kernel (NTK) framework for PINNs to establish global convergence guarantees for a broad class of linear operators. For nonlinear PDEs, we prove convergence to critical points via the Łojasiewicz inequality under the random feature model, eliminating the need for strong over-parameterization and encompassing both gradient flow and implicit gradient descent dynamics. Our results further reveal that the random feature model exhibits an implicit regularization effect, preventing parameter divergence to infinity. Theoretical findings are corroborated by numerical experiments, providing new insights into the training dynamics and robustness of neural network PDE solvers.

## 1 Introduction

Partial differential equations (PDEs) form the mathematical foundation for modeling phenomena across physics, engineering, and applied sciences. While linear PDEs are relatively well-understood, nonlinear PDEs, ubiquitous in modeling complex systems, pose significant analytical and computational challenges due to their lack of superposition principles and potential for solution singularities (Evans, 2022; Johnson, 2009). Recent advances in machine learning have introduced neural PDE solvers, such as physics-informed neural networks (Raissi et al., 2019) and the Deep Ritz method (E & Yu, 2018), as flexible alternatives to traditional numerical methods. These approaches have demonstrated empirical success in high-dimensional and nonlinear settings (Lawal et al., 2022; Karniadakis et al., 2021; Liao & Ming, 2021; Liu et al., 2023), but their theoretical convergence guarantees remain limited, especially for nonlinear PDEs.

Most existing convergence analyses for physics-informed neural networks are developed within the neural tangent kernel framework (Jacot et al., 2018; Li et al., 2020), which primarily provides guarantees for second-order linear PDEs using over-parameterized networks (Gao et al., 2023; Xu et al., 2024a;b). While it is commonly believed that NTK-based results could be extended to broader classes of linear PDEs, rigorous proofs beyond the second-order setting are still lacking. For the Deep Ritz method, convergence analyses typically rely on coercivity of the bilinear form and Rademacher complexity estimates (Duan et al., 2022; Jiao et al., 2024; Lu et al., 2021); however, these approaches are mostly confined to linear elliptic equations with convex energy functionals, and the extension to general variational problems remains underexplored. Crucially, neither framework currently offers provable convergence guarantees for solving nonlinear PDEs. In particular, when PINNs are used to solve equations with nonlinear differential operators, the associated NTK matrix evolves dynamically during training and, as shown in Bonfanti et al. (2024), fails to converge to a deterministic kernel in the infinite-width limit. For the Deep Ritz method, the non-convexity inherent in nonlinear PDEs further complicates the analysis. This theoretical gap poses a major challenge to our understanding of neural PDE solvers in nonlinear regimes.

In this work, we overcome the aforementioned theoretical limitations by establishing a systematic convergence theory for neural PDE solvers across both linear and nonlinear regimes. For linear PDEs, we extend the NTK framework to establish convergence guarantees for PINNs solving a broad

class of linear operators, surpassing results limited to second-order cases. For nonlinear PDEs, we introduce a new approach using the Łojasiewicz inequality (Haraux, 2012) to rigorously characterize optimization dynamics and guarantee convergence to critical points for important nonlinear cases. This provides the first convergence theory unifying PINNs and Deep Ritz methods in nonlinear settings. More precisely, the main contributions of this paper are as follows:

(i) We establish convergence to global minima for over-parameterized PINNs in solving a broad class of linear PDEs, thereby significantly extending existing NTK-based results that are limited to second-order cases (Theorem 1).

(ii) We provide a convergence framework for PINN and Deep Ritz solvers under both gradient flow and implicit gradient descent dynamics, assuming coercivity of the loss function (Proposition 1).

(iii) Under the random feature model, we prove convergence to critical points for both PINN and Deep Ritz solvers when applied to a wide range of PDEs, including all evolutionary equations and several fundamental classes of nonlinear PDEs (Theorems 3 and 4). Moreover, our analysis reveals an intrinsic regularization effect induced by the random feature model.

This paper is organized as follows. In Section 2, we review related works on machine learning-based PDE solvers and existing convergence analyses. Section 3 introduces the problem setting, and establishes general convergence results. Section 4 presents our main convergence results for solving nonlinear PDEs under different cases. Section 5 provides experimental evidence supporting our theoretical findings. Finally, Section 6 concludes the paper and discusses potential directions for future research. Technical proofs and supplementary materials are included in the appendix.

## 2 RELATED WORKS

**Machine learning PDE solvers.**   There are various machine learning-based solvers for PDEs, among which physics-informed neural networks (Raissi et al., 2019) and Deep Ritz method (E & Yu, 2018) are the most widely used. PINNs incorporate the PDE structure directly into the loss function, while Deep Ritz leverages the variational form of certain problems. Both methods have demonstrated remarkable empirical performance in solving a wide variety of nonlinear PDEs including, for example, the Allen–Cahn equation (Wight & Zhao, 2021) and Schrödinger equation (Qiu et al., 2025) across numerous applications (Chen et al., 2024; Tang et al., 2023; Savović et al., 2023). Despite their success, theoretical understanding of their convergence properties, particularly for nonlinear PDEs, remains limited and is an active area of ongoing research.

**Existing convergence analysis using NTK framework.**   The neural tangent kernel (NTK) framework, which approximates over-parameterized neural networks as linear models with an almost constant Gram matrix during training, underpins much of the existing convergence analysis (Jacot et al., 2018; Li et al., 2020). NTK was initially applied to study gradient descent in supervised learning settings (Du et al., 2019a; Luo & Yang, 2024; Du et al., 2019b), and has been extended to analyze the convergence of PINNs for second-order linear PDEs (Gao et al., 2023; Xu et al., 2024a;b). These results typically show that, for highly over-parameterized NTK-scaled neural networks, the training loss converges to zero with gradient-based optimization methods.

**Convergence analysis using Łojasiewicz inequality.**   The Łojasiewicz inequality (Haraux, 2012) is a fundamental analytical tool in the field of optimization, especially for studying the convergence properties of gradient-based algorithms (Bolte et al., 2007; Alaa & Pierre, 2013). Traditionally, it has been widely used to analyze the convergence in various non-convex and nonsmooth optimization problems (Schneider & Uschmajew, 2015; Attouch et al., 2010; Karimi et al., 2016). In recent years, the Łojasiewicz inequality has also been increasingly applied in the context of supervised learning (Forti et al., 2006; Li et al., 2023a). Researchers have leveraged this inequality to study the convergence behavior of machine learning algorithms, providing theoretical guarantees for global or local convergence under mild assumptions (Lee et al., 2016; Ahmadova, 2023).

## 3 MATHEMATICAL SETUP AND GENERAL SUPPORTING RESULTS

In this section, we present the basic mathematical setup and introduce both PINNs and Deep Ritz solvers. In Section 3.2, we establish two main results: first, a rigorous global convergence guarantee

for PINNs in solving a broad class of linear PDEs based on the NTK approach; second, a more general convergence result to critical points, which serves as the foundation for our subsequent analysis of nonlinear PDEs.

## 3.1 Problem setting

We consider a general class of partial differential equations defined on an open bounded domain $\Omega \subset \mathbb{R}^d$ with $d > 1$, taking the following form:

$$\begin{cases} \mathcal{L}u = f, & x \in \Omega, \\ \mathcal{B}u = g, & x \in \partial\Omega, \end{cases} \tag{1}$$

where $\mathcal{L}$ represents a differential operator that may be linear or nonlinear, and $f \in L^\infty(\Omega)$ denotes the source term. For evolutionary PDE, we adopt the convention where the first component of $x$ represents the temporal dimension while the remaining components correspond to spatial coordinates, thus naturally satisfying $d > 1$. The boundary conditions are encoded through the operator $\mathcal{B}$, which we specify as Robin-type: $\alpha u(x) + \beta \frac{\partial u}{\partial n}(x) = g(x)$ for $x \in \partial\Omega$, where $\alpha, \beta \in \mathbb{R}$ are not both zero, $g \in L^2(\partial\Omega)$ is the prescribed boundary data, and $\frac{\partial u}{\partial n}$ denotes the outward normal derivative.

In this work, we focus on two neural network-based approaches for solving PDEs: physics-informed neural networks (PINNs) and the Deep Ritz method. Both leverage the expressive power of deep networks to approximate the solution $u$. In PINNs, a neural network $u_\theta(x)$ parameterized by $\theta$ is trained by minimizing the composite loss:

$$\mathcal{J}_{\text{PINN}}(\theta) = \int_\Omega (\mathcal{L}u_\theta(x) - f(x))^2 \, \mathrm{d}x + \lambda \int_{\partial\Omega} (\mathcal{B}u_\theta(x) - g(x))^2 \, \mathrm{d}x, \tag{2}$$

where $\lambda \geq 0$ balances the PDE residual and boundary losses. The Deep Ritz method, applicable to PDEs with variational structure $\mathcal{E}(u)$, seeks a minimizer $u_\theta$ of the loss function:

$$\mathcal{J}_{\text{Ritz}}(\theta) = \mathcal{E}(u_\theta) + \lambda \int_{\partial\Omega} (\mathcal{B}u_\theta(x) - g(x))^2 \, \mathrm{d}x. \tag{3}$$

In our theoretical framework, we employ a two-layer neural network with $\tanh$ activation function to approximate the solution to Eq.(1). Specifically, the network takes the form:

$$u_\theta(x) = \sum\nolimits_{k=1}^m a_k \, \tanh(w_k^\mathsf{T} x + b_k), \tag{4}$$

where $\theta = \{(a_k, w_k, b_k)\}_{k=1}^m$ denotes all parameters, with $a_k \in \mathbb{R}, w_k \in \mathbb{R}^d$ and $b_k \in \mathbb{R}$. The $\tanh$ activation function is particularly well-suited for our analysis due to its analyticity and bounded derivatives. More crucially, it satisfies a key property (see Lemma 1) that, in combination with its other features, underpins our convergence theory. The following lemma is proved in Section B.

**Lemma 1** (Linear independence). *Let $m$ be a positive integer, and let $\alpha, \beta \in \mathbb{R}$ be not both zero. Given real numbers $p_1, \ldots, p_m$ such that $p_i \neq \pm p_j$ for $1 \leq i \neq j \leq m$, and $q_1, \ldots, q_m \in \mathbb{R}$, the functions $\alpha \tanh(p_1 t + q_1) + \beta \tanh'(p_1 t + q_1), \ldots, \alpha \tanh(p_m t + q_m) + \beta \tanh'(p_m t + q_m)$ are linearly independent over $\mathbb{R}$.*

**Remark 1** (On the choice of activation functions). *Our analysis relies on three key properties of the activation function: bounded derivatives, analyticity, and the linear independence property stated in Lemma 1. These hold for a broad class of analytic activations, such as $\text{sigmoid}$ and $\arctan$. The theoretical framework can be readily extended to any activation function satisfying these conditions.*

## 3.2 General convergence results

This subsection presents two convergence results for neural PDE solvers. While the NTK framework guarantees global convergence for solving most linear PDEs, it can not extend to nonlinear cases. This limitation motivates our alternative approach based on Łojasiewicz analysis, which establishes critical point convergence beyond the linear setting.

### 3.2.1 NTK-based convergence for most linear PDEs

While prior PINN convergence theories have mainly focused on second-order linear PDEs, such as the heat equation, we extend existing analytical techniques to establish the first global convergence guarantees for solving a broad class of linear PDEs. We begin by introducing some notations.

**Notation:** Let $\xi = (\xi_1, \ldots, \xi_d) \in \mathbb{N}^d$ be a $d$-dimensional multi-index, where $\mathbb{N}$ denotes the set of non-negative integers. Given a vector $x = (x_1, \ldots, x_d)^\mathsf{T} \in \mathbb{R}^d$, we define the $\xi$-th power of $x$ as $x^\xi := \prod_{i=1}^d x_i^{\xi_i}$. For a sufficiently smooth function $u : \mathbb{R}^d \to \mathbb{R}$, its $\xi$-th partial derivative is denoted by $\partial^\xi u := \frac{\partial^{|\xi|} u}{\partial x_1^{\xi_1} \cdots \partial x_d^{\xi_d}}$, where $|\xi| := \sum_{i=1}^d \xi_i$ represents the order of the derivative. For two positive functions $f_1(n)$ and $f_2(n)$, we use $f_1(n) = \mathcal{O}(f_2(n))$, $f_2(n) = \Omega(f_1(n))$, or $f_1(n) \lesssim f_2(n)$ to indicate that $f_1(n) \leq C f_2(n)$, where $C$ is a universal constant. If we further omit some logarithmic terms with the existence of polynomial terms, we adopt $f_1(n) = \widetilde{\mathcal{O}}(f_2(n))$ and $f_2(n) = \widetilde{\Omega}(f_1(n))$.

**Definition 1** (Admissible linear operators). *Let $\mathcal{L}$ be a linear differential operator of the form $\mathcal{L}u(x) = \sum_{k=0}^\infty \sum_{|\xi|=k} c_\xi(x) \partial^\xi u$, where only finitely many coefficients $c_\xi$ are nonzero. We require that all nonzero $c_\xi \in L^\infty(\Omega)$, and that there exists a maximal multi-index $\tilde{\xi}$ such that $c_{\tilde{\xi}} \neq 0$ and $|\tilde{\xi}| > |\xi|$ for all other $\xi$ with $c_\xi \neq 0$.*

Under the neural tangent kernel framework, we use a rescaled two-layer neural network of the form:

$$u_\theta(x) = \frac{1}{\sqrt{m}} \sum_{k=1}^m a_k \tanh(w_k^T x + b_k), \tag{5}$$

where the scaling factor $\frac{1}{\sqrt{m}}$ ensures proper normalization for theoretical analysis. Within the PINN framework, the empirical loss combines PDE residual and boundary terms on collocation points as follows,

$$\mathcal{J}_{\text{emp}}(\theta) = \frac{1}{n_1} \sum_{i=1}^{n_1} \frac{1}{2} \left| \mathcal{L}u_\theta\left(x_i^{(1)}\right) - f\left(x_i^{(1)}\right) \right|^2 + \frac{\lambda}{n_2} \sum_{j=1}^{n_2} \frac{1}{2} \left| \mathcal{B}u_\theta\left(x_j^{(2)}\right) - g\left(x_j^{(2)}\right) \right|^2, \tag{6}$$

with collocation points $\{x_i^{(1)}\}_{i=1}^{n_1} \subset \Omega$ and $\{x_j^{(2)}\}_{j=1}^{n_2} \subset \partial\Omega$. Under gradient flow training, we show that $\mathcal{J}_{\text{emp}}(\theta(t))$ converges to the global minimum of the empirical loss if $\mathcal{L}$ is admissible.

**Theorem 1** (Convergence for admissible linear PDEs). *Assume that the linear differential operator $\mathcal{L}$ is admissible. Consider the gradient flow dynamics for Eq.(6), $\frac{d\theta(t)}{dt} = -\nabla \mathcal{J}_{emp}(\theta)$. Given training samples $\{x_i^{(1)}\}_{i=1}^{n_1} \subset \Omega$ and $\{x_j^{(2)}\}_{j=1}^{n_2} \subset \partial\Omega$, initialize the parameters in Eq.(5) as $a_k \sim Unif\{-1, 1\}$, $w_k \sim \mathcal{N}(0, \mathbf{I}_d)$, $b_k \sim \mathcal{N}(0, 1)$ i.i.d. Then, with probability at least $1 - \delta$,*

$$\mathcal{J}_{emp}(\theta(t)) \leq \exp\left( -(\lambda_0 + \tilde{\lambda}_0)t \right) \mathcal{J}_{emp}(\theta(0)), \quad \forall t \geq 0,$$

*provided that $m = \widetilde{\Omega}\left( \frac{1}{\left(\lambda_0 + \tilde{\lambda}_0\right)^2} d^{4|\tilde{\xi}|} \left( \log\left( \frac{n_1 + n_2}{\delta} \right) \right)^{4|\tilde{\xi}|} \frac{d^3}{\min\{\lambda_0^2, \tilde{\lambda}_0^2\}} \right)$.*

**Remark 2.** *The proof of this theorem is provided in Section G. As shown in the proof, $\lambda_0, \tilde{\lambda}_0$ are actually the minimum eigenvalues of the Gram matrices respectively.*

**Remark 3** (Various extensions). *Theorem 1 can extend to several broader contexts:*

*(1)* Other training dynamics: *Leveraging the NTK analysis, the theorem applies to gradient descent with sufficiently small step sizes, implicit gradient descent, and other initialization schemes. Possible extensions to SGD are discussed in Section J.1.*

*(2)* More complex network architectures: *The proof strategy adapts to deeper networks, following extensions of NTK theory in the supervised learning (Du et al., 2019b); see also Section H.1.*

*(3)* Broader classes of linear operators: *While we focus on admissible linear operators, similar techniques apply to a broader class of linear operators. Further details are omitted for brevity.*

### 3.2.2 FROM LINEAR TO NONLINEAR PDES

Our NTK-based convergence theory for linear PDEs relies on the near-constancy of the Gram matrix during training, a property that does not hold for nonlinear PDEs as proved in Bonfanti et al. (2024). To illustrate this point more intuitively, we present a numerical experiment. Consider the viscous Burgers' equation:

$$\begin{cases} u_t + u u_x = \frac{0.01}{\pi} u_{xx}, & t \in (0, 1), \ x \in (-1, 1), \\ u(0, x) = -\sin(\pi x), & x \in (-1, 1), \\ u(t, -1) = u(t, 1) = 0, & t \in (0, 1). \end{cases} \tag{7}$$

We demonstrate the failure of NTK theory for Burgers' equation using a neural network with architecture described in Eq.(5) (with width $m = 1000$); detailed experimental settings are provided in Section I.2. For simplicity, we train only the outer-layer parameters $a = (a_1, \ldots, a_m)^\top$ using implicit gradient descent (see Section I.1 for algorithmic details). During training, we track the evolution of two NTK matrices:

(i) The interior NTK matrix $K_\Omega(t) = (K_{ij})$ is computed from the derivatives of the PDE residual evaluated at 100 interior collocation points, $X^{(1)} = \{(t_k^{(1)}, x_k^{(1)})\}_{k=1}^{100} \subset (0,1) \times (-1,1)$. Specifically, $K_{ij} = \left\langle \partial_a r_\theta(t_i^{(1)}, x_i^{(1)}), \partial_a r_\theta(t_j^{(1)}, x_j^{(1)}) \right\rangle$, $i, j = 1, \ldots, 100$, where the PDE residual is $r_\theta(t, x) = \left(\partial_t u_\theta + u_\theta \partial_x u_\theta - \frac{0.01}{\pi} \partial_{xx} u_\theta\right)(t, x)$.

(ii) The boundary NTK matrix $K_{\partial\Omega}(t) = (\tilde{K}_{ij})$ is computed from the derivatives of the network output at 20 sampled boundary points, $X^{(2)} = \{(t_k^{(2)}, x_k^{(2)})\}_{k=1}^{20}$. Specifically, $\tilde{K}_{ij} = \left\langle \partial_a u_\theta(t_i^{(2)}, x_i^{(2)}), \partial_a u_\theta(t_j^{(2)}, x_j^{(2)}) \right\rangle$, $i, j = 1, \ldots, 20$.

The IGD algorithm is run for 100 iterations with a step size of 0.5, where each inner optimization problem is approximately solved by applying the L-BFGS optimizer for 10 steps. As shown in Figure 1, $K_\Omega(t)$ undergoes significant changes from its initial state within just a few iterations, whereas $K_{\partial\Omega}(t)$ remains nearly unchanged, as the boundary operator is linear.

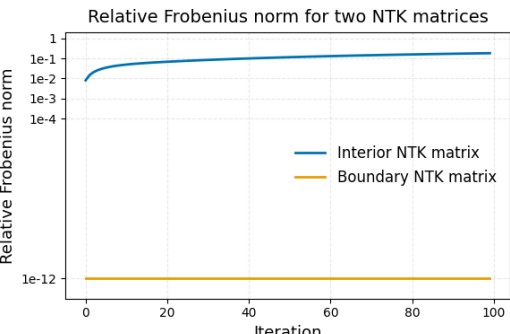

Figure 1: Evolution of relative Frobenius norm for two NTK matrices.

Thus, new mathematical tools are required to analyze the convergence of PINNs and Deep Ritz methods for solving nonlinear PDEs. Given that strong overparameterization is difficult to verify in practice, we forgo this assumption and instead employ the Łojasiewicz inequality to establish convergence to critical points, albeit with weaker guarantees.

### 3.2.3 CONVERGENCE UNDER COERCIVITY

We now present a general result showing that coercivity of the loss function implies convergence. In the subsequent section, we demonstrate the coercivity of the loss function, with a focus on those arising in nonlinear PDEs, thus allowing us to apply the general convergence result obtained here.

**Definition 2** (Coercivity). *A function $\mathcal{J}(\theta)$ is said to be coercive if $\lim_{\|\theta\| \to +\infty} \mathcal{J}(\theta) = +\infty$.*

The coercivity help to ensure boundedness of minimizing sequences, a crucial property for convergence analysis. We next introduce the Łojasiewicz inequality, which is fundamental to our analysis.

**Theorem 2** (Łojasiewicz inequality, Theorem 1.1 in Haraux (2012)). *Let $U$ be an open subset of $\mathbb{R}^N$ and $F : U \to \mathbb{R}$ be a real analytic function. Then for any $x$ in $U$ such that $\nabla F(x) = 0$, there exist a neighbourhood $W$ of $x$ and a real number $\epsilon \in (0, \frac{1}{2}]$ for which $\forall y \in W, |F(y) - F(x)|^{1-\epsilon} \leq \|\nabla F(y)\|$. We call $\epsilon$ the Łojasiewicz exponent of $F$ at $x$.*

We denote the loss function of either PINN Eq.(2) or Deep Ritz Eq.(3) as

$$\mathcal{J} = residual\ (variational)\ term + \lambda \int_{\partial\Omega} (\mathcal{B}(u_\theta) - g)^2 \, \mathrm{d}x. \tag{8}$$

For this loss function, we consider two types of training dynamics. The first is gradient flow,

$$\theta'(t) = -\nabla \mathcal{J}(\theta). \tag{9}$$

which provides a continuous-time perspective on optimization. The second is implicit gradient descent (IGD), which we disscuss in detail in Section I.1,

$$\theta^{k+1} = \theta^k - \eta \nabla \mathcal{J}(\theta^{k+1}), \quad k = 0, 1, \dots \tag{10}$$

where $\eta$ is the step size. As introduced in Li et al. (2023b), IGD enjoys greater stability compared to standard gradient descent and is particularly well-suited for multi-scale problems. We now establish convergence results for both dynamics under coercivity. The proof of the following proposition is provided in Section C.

**Proposition 1** (Convergence under coercivity). *Suppose $\mathcal{J}(\theta)$ is coercive in $\theta$. Then:*

*(a) The solution $\theta(t)$ to the gradient flow Eq.(9) converges to a critical point $\theta^*$ of $\mathcal{J}(\theta)$ as $t \to \infty$.*

*(b) The sequence $\{\theta^k\}$ generated by Eq.(10) converges to a critical point $\theta^*$ of $\mathcal{J}(\theta)$ as $k \to \infty$.*

*Furthermore, let $\epsilon$ denote the Łojasiewicz exponent of $\mathcal{J}(\theta)$ at $\theta^*$. The convergence rates are as follows:*

*(i) If $\epsilon \in (0, \frac{1}{2})$, then for some $C > 0$ and integer $k_0$,*

$$\|\theta(t) - \theta^*\|_2 \le C\, t^{-\frac{\epsilon}{1-2\epsilon}}, \ \forall t > 0; \quad \|\theta^k - \theta^*\|_2 \le C\, (k\eta)^{-\frac{\epsilon}{1-2\epsilon}}, \ \forall k > k_0.$$

*(ii) If $\epsilon = \frac{1}{2}$, then for some $C > 0$ and integer $k_0$,*

$$\|\theta(t) - \theta^*\|_2 \le C\, e^{-t}, \ \forall t > 0; \quad \|\theta^k - \theta^*\|_2 \le C\, e^{-k\eta}, \ \forall k > k_0.$$

**Remark 4.** *The convergence rate deteriorates as $\epsilon$ approaches zero. A similar phenomenon is observed in NTK-based analyses: when the minimum eigenvalue of the NTK matrix is close to zero, convergence also slows down.*

**Remark 5.** *Our convergence results hold for gradient descent with proper step size choices (omitted for brevity), while implicit gradient descent offers additional advantages as it maintains unconditional stability and better preserves the solution structure throughout training.*

**Remark 6** (Advantages of implicit regularization). *While explicit $L^2$ regularization, adding a term such as $\gamma\|\theta\|_2^2$ to the loss, ensures coercivity, modern PDE solvers like PINNs and the Deep Ritz method predominantly rely on implicit regularization induced by gradient-based optimization algorithms. This offers several key advantages: it naturally promotes low-norm solutions without the need for careful tuning of $\gamma$, preserves the physical interpretability of the loss, and avoids artificially restricting the solution space. Importantly, implicit regularization adapts robustly to multiscale features (such as sharp gradients and boundary layers) that are common in practical PDE problems. Extensive empirical results demonstrate that this implicit effect often yields a better trade-off between training stability and solution accuracy across a wide range of benchmarks.*

## 4 CONVERGENCE FOR SOLVING NONLINEAR PDES

In this section, we present a rigorous coercivity analysis of the loss function for DNN-based solvers, including both PINNs and the Deep Ritz method. By Proposition 1, establishing coercivity is crucial for guaranteeing the convergence when solving a broad class of PDEs, especially nonlinear ones.

### 4.1 RANDOM FEATURE MODEL

Even for two-layer neural networks Eq.(4), the loss function for complex nonlinear PDEs can be highly intricate. As a first step, we focus on the random feature model (Chen et al., 2023). In this setting, the network structure remains as in Eq.(4), but the inner-layer parameters $w_k = (w_{k,1}, \dots, w_{k,d})^\mathsf{T} \in \mathbb{R}^d$ and $b_k \in \mathbb{R}$ are randomly initialized and kept fixed during training; only the outer-layer coefficients $a = (a_1, \cdots, a_m)^\mathsf{T}$ are trainable.

In typical physics-informed learning methods, the loss function naturally admits the decomposition $\mathcal{J}(a) = \mathcal{J}_\Omega(a) + \lambda\, \mathcal{J}_{\partial\Omega}(a)$, where $\mathcal{J}_\Omega(a)$ enforces either the PDE residual (for PINNs) or the variational functional (for the Deep Ritz method) in the interior of the domain, and $\mathcal{J}_{\partial\Omega}(a) = \int_{\partial\Omega} (\mathcal{B}(u_\theta) - g)^2 \, \mathrm{d}x$ imposes the boundary conditions. Based on this decomposition, we reveal two distinct mechanisms by which $\mathcal{J}$ exhibits coercivity with respect to $a$:

**Case 1 (Boundary-induced coercivity):** Under mild assumptions, the boundary term $\mathcal{J}_{\partial\Omega}(a)$ dominates in such a way that there exists a constant $C > 0$ such that $\mathcal{J}_{\partial\Omega}(a) \geq C\|a\|_2^2$. So the loss function $\mathcal{J}(a)$ is coercive with respect to $a$.

**Case 2 (Interior-induced coercivity):** For certain second-order nonlinear PDEs, the interior term $\mathcal{J}_{\Omega}(a)$ provides coercivity, i.e., there exists $C > 0$ such that $\mathcal{J}_{\Omega}(a) \geq C\|a\|_2^2$.

The following subsections establish precise sufficient conditions for each case, thereby covering most practical PDEs encountered in applications.

## 4.2 BOUNDARY-INDUCED COERCIVITY

We begin by specifying our geometric assumptions on the domain $\Omega$. The key requirement is that the boundary $\partial\Omega$ contains a sufficiently regular portion that can be transformed into a flat segment. Formally, we make the following assumption:

**Assumption 1** (Local flat boundary). *There exists an invertible affine transformation* $\mathrm{Aff} : x \mapsto Ax + w_0$ *such that the transformed domain* $\tilde{\Omega} = \mathrm{Aff}(\Omega)$ *satisfies:*

*(i) local flatness: for some point* $y^* \in \partial\tilde{\Omega}$ *and* $r > 0$, $\partial\tilde{\Omega} \cap B(y^*, r) = \{y \in B(y^*, r) : y_d = \gamma\}$, *where* $B(y^*, r)$ *denotes the open ball of radius* $r$ *centered at* $y^*$ *in* $\mathbb{R}^d$, $y_d$ *is the* $d$-*th coordinate of* $y$, *and* $\gamma$ *is a constant.*

*(ii) non-degeneracy: the flat boundary portion has positive* $(d$-$1)$-*dimensional measure, i.e.,* $\lambda_{d-1}(\partial\tilde{\Omega} \cap B(y^*, r)) > 0$. *We denote* $\Gamma := \mathrm{Aff}^{-1}(\partial\tilde{\Omega} \cap B(y^*, r))$ *as the corresponding boundary portion in the original coordinates.*

**Remark 7.** *This assumption is naturally satisfied for evolutionary PDEs, where* $\Gamma$ *can be taken as the initial time slice* $\{t = 0\}$. *In practical settings, local flat boundaries are common. For instance, they naturally appear in domains with piecewise smooth or polyhedral boundaries. Therefore, Assumption 1 introduces only a weak and broadly applicable geometric condition.*

For notational simplicity, we will work in coordinates where $\mathrm{Aff}$ is the identity transformation, i.e., $A = \mathrm{Id} \in \mathbb{R}^{d \times d}, w_0 = 0 \in \mathbb{R}^d$. This does not affect the generality of our results due to the affine invariance of the coercivity property. To establish coercivity, we first characterize a class of well-behaved neural network inner-layer parameters that guarantee desirable properties.

**Definition 3** (Admissible inner-layer parameters). *Denote the first* $d-1$ *coordinates of* $w_k$ *as* $\tilde{w}_k = (w_{k,1}, \ldots, w_{k,d-1})^{\mathsf{T}}$. *The parameter set* $\{(w_k, b_k)\}_{k=1}^m$ *is called admissible inner-layer parameters if the following two conditions are satisfied:*

*(i) distinct directional components:* $\tilde{w}_i \neq \pm\tilde{w}_j$ *for any* $1 \leq i < j \leq m$;

*(ii) non-degenerate normal components:* $w_{i,d} \neq 0$ *for any* $1 \leq i \leq m$.

We now establish the coercivity of the loss function $\mathcal{J}(a)$ under the admissible inner-layer parameters condition, as formalized in the following result proved in Section D.1.

**Proposition 2** (Boundary linear independence). *For admissible inner-layer parameters, the functions*

$$\alpha \tanh(w_1^{\mathsf{T}} x + b_1) + \beta w_{1,d} \tanh'(w_1^{\mathsf{T}} x + b_1), \ldots, \alpha \tanh(w_m^{\mathsf{T}} x + b_m) + \beta w_{m,d} \tanh'(w_m^{\mathsf{T}} x + b_m)$$

*are linearly independent in* $L^2(\Gamma)$. *Furthermore, recall that* $u_\theta = \sum_{k=1}^m a_k \tanh(w_k^{\mathsf{T}} x + b_k)$, *then there exits a constant* $C > 0$ *such that* $\|a\|_2 \leq C \left\|\alpha u_\theta + \beta \frac{\partial u_\theta}{\partial n}\right\|_{L^2(\Gamma)}$.

The significance of this result lies at the core of our analysis. By astutely exploiting the linear independence of these functions, we are able to rigorously bound $\|a\|_2$ using the boundary data. Based on this estimate, we are able to establish the following convergence theorem.

**Random Initialization** Inner-layer parameters $\{(w_k, b_k)\}_{k=1}^m$ are randomly initialized according to the following rule: $w_i \sim \mathcal{N}(0, \mathbf{I}_d)$ i.i.d. ; $b_i \sim \mathcal{N}(0, 1)$ i.i.d. for $1 \leq i \leq m$.

**Theorem 3** (Almost sure convergence via admissible initialization). *Under Assumption 1, regardless of the specific form of the differential operator* $\mathcal{L}$ *in the PDE, we can initialize the inner parameters* $\{(w_k, b_k)\}_{k=1}^m$ *with probability 1 such that (i)* $\mathcal{J}$ *is coercive with respect to* $a$; *and (ii) all convergence results of Proposition 1 hold.*

*Proof.* By construction, for randomly initialized inner parameters, the admissibility condition of Definition 3 is satisfied almost surely. Then, for almost surely inner parameters, by Proposition 2, there exists $C > 0$ such that $\|a\|_2 \leq C \left\|\alpha u_\theta + \beta \frac{\partial u_\theta}{\partial n}\right\|_{L^2(\Gamma)}$, where $u_\theta = \sum_{k=1}^m a_k \tanh(w_k^\top x + b_k)$. This directly implies that $\mathcal{J}(a) \to +\infty$ as $\|a\| \to +\infty$, i.e., $\mathcal{J}$ is coercive with respect to $a$. Consequently, by invoking Proposition 1, we conclude that, for almost every realization of the inner parameters, the convergence results therein hold. This completes the proof. $\square$

This theorem ensures that random initialization almost surely yields admissible parameters satisfying the coercivity condition. As a result, the convergence results apply generically to both PINNs and the Deep Ritz method, regardless of the choice of differential operator in the PDE. A similar statement for the empirical loss is discussed in Section D.2.

**Remark 8.** *Extensions to deeper networks and the challenges arising when inner-layer parameters are also trainable are discussed in Section H.2. Further discussion of the possibilities and difficulties of extending these results to SGD appears in Section J.2.*

### 4.3 INTERIOR-INDUCED COERCIVITY FOR SPECIFIC PDES

We now discuss in detail the coercivity of the interior loss introduced in Case 2 above; analogous results for Deep Ritz solvers are given in Section F. Consider the following prototypical nonlinear operators with homogeneous Dirichlet conditions ($u|_{\partial\Omega} = 0$):

$$
\begin{aligned}
(i) \quad & -\operatorname{div}(|\nabla u|^{p-2}\nabla u) + q(x)u + h(u), \quad p \geq 2,\ q \geq 0,\ h(u)u \geq 0; \\
(ii) \quad & -\operatorname{div}((1+u^2)\nabla u) + q(x)u + h(u), \quad q \geq 0,\ h(u)u \geq 0.
\end{aligned}
\tag{11}
$$

To strictly enforce homogeneous Dirichlet conditions, we multiply the neural network by a cutoff function $\varphi(x)$ that vanishes on $\partial\Omega$. Let $\varphi(x)$ be a smooth function such that $0 \leq \varphi(x) \leq 1$ on $\Omega$, $\varphi(x) = 0$ on $\partial\Omega$ and $\varphi(x) \equiv 1$ on some open set $U \subset \Omega$. The modified ansatz $\tilde{u}_\theta(x) := \varphi(x)u_\theta(x)$ automatically satisfies the boundary conditions, allowing us to focus on learning the interior.

**Proposition 3** (Interior $L^2$ control). *For operators in Eq.(11), there exists $C > 0$ such that*

$$
\|u\|_{L^2(U)} \leq C(\|\mathcal{L}\tilde{u} - f\|_{L^2(\Omega)} + \|f\|_{L^2(\Omega)}).
$$

The proof of this proposition is provided in Section E. This stability estimate directly enables coercivity through interior terms alone, complementing our boundary-based results. By applying the same techniques as in the previous section, we can conclude the following theorem.

**Theorem 4** (Almost sure convergence for PINNs). *Using PINNs to solve $\mathcal{L}u = f$ with homogeneous Dirichlet boundary condition, where $\mathcal{L}$ is defined as in Eq.(11), we can initialize the inner parameters $\{(w_k, b_k)\}_{k=1}^m$ with probability 1 such that*

*(i) $w_i \neq \pm w_j$ for $1 \leq i < j \leq m$, then $\{\tanh(w_k^\top x + b_k)\}_{k=1}^m$ are linearly independent in $L^2(U)$;*

*(ii) the loss function $\mathcal{J}_{PINN}$ defined in Eq.(2) is coercive about $a$;*

*(iii) all convergence results of Proposition 1 hold.*

*Proof.* Fix any choice of inner-layer parameters $\{(w_k, b_k)\}_{k=1}^m$. By Proposition 3, we have

$$
\|u_\theta\|_{L^2(U)} \leq C(\|\mathcal{L}\tilde{u} - f\|_{L^2(\Omega)} + \|f\|_{L^2(\Omega)}) \leq C(\mathcal{J}(a) + \|f\|_{L^2(\Omega)}).
$$

According to Lemma 1, the functions $\{\tanh(w_k^\top x + b_k)\}_{k=1}^m$ are linearly independent in $L^2(U)$ provided $w_i \neq \pm w_j$ for all $i \neq j$. Standard linear algebra then yields

$$
\|a\|_2 \leq C\|u_\theta\|_{L^2(U)} \leq C(\mathcal{J}(a) + \|f\|_{L^2(\Omega)}).
$$

Since the set of parameters $\{(w_k, b_k)\}_{k=1}^m$ with $w_i \neq \pm w_j$ for all $i \neq j$ has full measure under random initialization, this estimate holds almost surely. Thus, the loss $\mathcal{J}_{PINN}$ is coercive with respect to $a$ for almost every initialization. The convergence result of Proposition 1 then follows. $\square$

**Implicit regularization of random feature model.** Establishing coercivity is crucial for our analysis. The direct application of Łojasiewicz inequality implies that optimization dynamics for all trainable parameters will either converge to a critical point or diverge to infinity. Coercivity rules out the latter by guaranteeing boundedness of the parameter sequence. Importantly, even in the presence of highly complex loss landscapes as encountered in PINNs or Deep Ritz frameworks, our results establish that, under the assumptions of Theorem 3 or Theorem 4, the random feature model provides inherent implicit regularization: both gradient flow and implicit gradient descent dynamics remain constrained within a bounded region, precluding divergence of the parameters or their gradients. Thus, no additional regularization technique is needed to prevent parameter or gradient explosion when using random feature models, even in these challenging settings. These results underscore the robustness of the random feature model in maintaining well-behaved optimization trajectories solely due to its intrinsic structural properties under mild conditions.

## 5 NUMERICAL EXPERIMENTS

As discussed earlier, time-dependent PDEs naturally satisfy Assumption 1. To validate Theorem 3, we test three representative time-dependent equations: the Burgers', Allen–Cahn, and Fisher–KPP equations. Detailed results for the Allen–Cahn and Fisher–KPP equations are given in Section I.3.1 and Section I.3.2, respectively. Here, we focus on the convergence behavior of the random feature model within the PINN framework for the Burgers' equation Eq.(7). A comprehensive summary of experimental hyperparameters is provided in Section I.2. Notably, our results do not rely on network over-parameterization. We train a network with $m = 100$ hidden units using implicit gradient descent (IGD) with a step size of $0.5$ for sufficient iterations, employing $10{,}000$ interior collocation points and $100$ boundary points. The final $\ell_2$-norm of the loss gradient is $1.13 \times 10^{-3}$, confirming convergence to a critical point even with a comparatively large step size.

Next, we consider the second equation in Eq.(11), which is also solved using the PINN framework with the random feature model, trained by IGD for sufficient iterations (see Section I.4 for further details). Table 1 summarizes the $\ell_2$-norm of the loss gradient with respect to the model parameters after training from different random initializations.

Table 1: Norm of the loss gradient with respect to $a$ after training from different initializations.

| Initialization | $\|\nabla \mathcal{J}(a)\|_2$ |
|---|---|
| Xavier normal for $w_k$, Xavier uniform for $a_k$ | $2.44 \times 10^{-4}$ |
| Standard normal for $w_k$, uniform $[-1, 1]$ for $a_k$ | $1.45 \times 10^{-4}$ |
| LeCun normal initialization for both $w_k$ and $a_k$ | $1.37 \times 10^{-4}$ |

These results consistently demonstrate a small loss gradient norm, further supporting convergence to a critical point as established in Theorem 4.

## 6 CONCLUSION

In this paper, we develop a unified convergence analysis for neural network-based PDE solvers, encompassing both linear and nonlinear equations. Leveraging the neural tangent kernel framework and the Łojasiewicz inequality within the random feature model, we establish rigorous convergence guarantees and highlight the intrinsic implicit regularization effect of the random feature approach. Our theoretical results show that both gradient flow and implicit gradient descent can achieve reliable convergence under mild conditions, even for nonlinear problems. While our current analysis for nonlinear PDEs focuses on random feature models, future work will seek to extend these results to fully-trainable architectures under suitable assumptions. We also intend to investigate optimization dynamics near saddle points and clarify the conditions distinguishing convergence to local versus global minima. Pursuing these directions will further strengthen the theoretical foundations and enhance the practical reliability of neural network-based PDE solvers.

ETHICS STATEMENT

This paper adheres to the ICLR Code of Ethics, as acknowledged and committed to by all authors.

REPRODUCIBILITY STATEMENT

All theoretical results in this paper are supported by complete proofs provided in the appendix. Detailed descriptions of all experimental setups—for both the main text and the appendix—are also included in the appendix to ensure reproducibility. Moreover, we provide the full source code for all experiments to further facilitate verification and reproduction of our results.

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

## A    The Use of Large Language Models (LLMs)

We used a large language model (GPT-4) to polish the language of the manuscript. Specifically, we drafted the initial versions of all sections ourselves, and then employed GPT-4 to refine the wording and clarity of selected passages—primarily introductory and expository paragraphs. The model did not contribute to research ideation, methodology, experiments, analyses, or conclusions.

## B    Proof of Lemma 1

We provide the proof of Lemma 1, which establishes an important property of the $\tanh$ activation function. More importantly, this ensures that our convergence results apply to neural networks with $\tanh$ activation, whether used in PINN or Deep Ritz solvers.

*Proof.* We first note that the $\tanh$ function is an odd function and $\tanh' = 1 - \tanh^2$ is an even function. So without loss of generality, we can assume that $p_1, \ldots, p_m$ are distinct positive numbers, otherwise, we replace $\tanh(p_r t + q_r)$ by $-\tanh(-p_r t - q_r)$ and $\tanh'(p_r t + q_r)$ by $\tanh'(-p_r t - q_r)$. We can also assume that $p_1 < p_2 < \cdots < p_m$. We divide the proof into two cases according to whether $\beta = 0$ or $\beta \neq 0$.

**Case 1:** $\beta = 0$. Given any positive integer $m$, for any set of $m$ real numbers

$$\{p_i : p_i \neq \pm p_j, \forall 1 \leq i < j \leq m\}$$

and any real numbers $q_1, \ldots, q_m$, we need to prove that the functions

$$\tanh(p_1 t + q_1), \cdots, \tanh(p_m t + q_m)$$

are linear independent.

Take $c_1, \ldots, c_m$ to be real numbers such that

$$c_1 \tanh(p_1 t + q_1) + \cdots + c_m \tanh(p_m t + q_m) = 0, \quad \forall t \in \mathbb{R}. \tag{12}$$

In the above equation, letting $t \to \infty$ and noting that all $p_r$ are positive real numbers, we obtain

$$c_1 + \cdots + c_m = 0. \tag{13}$$

Substituting $\tanh(t) = \frac{e^{2t} - 1}{e^{2t} + 1}$ into Eq.(12), we obtain

$$\sum_{k=1}^{m} c_k \frac{e^{2(p_k t + q_k)} - 1}{e^{2(p_k t + q_k)} + 1} = 0, \quad \forall t \in \mathbb{R}.$$

Multiplying both sides of the above equality by $\prod_{l=1}^{m}(e^{2(p_l t + q_l)} + 1)$, we have

$$\sum_{k=1}^{m} c_k (e^{2(p_k t + q_k)} - 1) \prod_{l=1, l \neq k}^{m} (e^{2(p_l t + q_l)} + 1) = 0, \quad \forall t \in \mathbb{R}. \tag{14}$$

In fact, each term in the above expression can be written in the following form:

$$\tilde{c}_K e^{2 \sum_{k \in K} (p_k t + q_k)},$$

where $K$ is a subset of $\{1, \ldots, m\}$ and $\tilde{c}_K$ is a constant.

Let us focus on one of these terms in particular, $e^{2(p_1 t + q_1)}$. By observing Eq.(14), we see that the coefficient in front of this term is $c_1 - \sum_{l \neq 1} c_l$. Moreover, since $p_1 < p_2 < \ldots < p_m$, we know that for any nonempty $\{1\} \neq K \subset \{1, \ldots, m\}$, $p_m < \sum_{k \in K} p_k$.

Thus, we can conclude that the coefficient $c_1 - \sum_{l \neq 1} c_l$ in front of $e^{2(p_1 t + q_1)}$ is non-zero. Otherwise, there exist constants $\alpha_K$ such that

$$e^{2(p_1 t + q_1)} = \sum_{K \neq \{1\}} \alpha_K e^{2 \sum_{k \in K} (p_k t + q_k)},$$

which contradicts the fact that for any $n$, the set $e^{a_i t}, a_i \neq a_j \ if \ i \neq j\}_{i=1}^{n}$ is linear independent.

Combining the above arguments, we can conclude that $\sum_{l=1}^{m} c_l = 0$ and $c_1 - \sum_{l \neq 1} c_l = 0$, thereby $c_1 = 0$. Following the same reasoning, we can similarly obtain $c_2 = 0, \ldots, c_m = 0$. Thus, this case is proved.

**Case 2:** $\beta \neq 0$. The underlying idea of the proof remains unchanged, but a more detailed treatment of the coefficient in front of $e^{2(p_1 t + q_1)}$ is required. We need to prove that the functions

$$\alpha \tanh(p_1 t + q_1) + \beta \tanh'(p_1 t + q_1), \ \ldots, \ \alpha \tanh(p_m t + q_m) + \beta \tanh'(p_m t + q_m)$$

are linear independent. Let us assume $\beta \neq -\frac{1}{2}$. If this is not the case, we can multiply $\alpha$ and $\beta$ by a common factor to arrange it so.

Take $c_1, \ldots, c_m$ to be real numbers such that

$$\sum_{k=1}^{m} c_k \left( \alpha \tanh(p_k t + q_k) + \beta \tanh'(p_k t + q_k) \right) = 0, \quad \forall t \in \mathbb{R}. \tag{15}$$

In the above equation, letting $t \to \infty$ and noting that all $p_r$ are positive real numbers, we obtain

$$c_1 + \cdots + c_m = 0. \tag{16}$$

Substituting $\tanh(t) = \frac{e^{2t}-1}{e^{2t}+1}$ and $\tanh'(t) = \frac{4e^{2t}}{(e^{2t}+1)^2}$ into Eq.(15), we obtain

$$\sum_{k=1}^{m} c_k \left[ \alpha \frac{e^{2(p_k t + q_k)} - 1}{e^{2(p_k t + q_k)} + 1} + \beta \frac{4e^{2(p_k t + q_k)}}{(e^{2(p_k t + q_k)} + 1)^2} \right] = 0, \quad \forall t \in \mathbb{R}.$$

Multiplying both sides of the above equality by $\prod_{l=1}^{m}(e^{2(p_l t + q_l)} + 1)^2$, we have

$$\sum_{k=1}^{m} c_k \left[ \alpha \left( e^{4(p_k t + q_k)} - 1 \right) + 4\beta e^{2(p_k t + q_k)} \right] \prod_{l=1, l \neq k}^{m} (e^{2(p_l t + q_l)} + 1)^2 = 0, \quad \forall t \in \mathbb{R}. \qquad (17)$$

In fact, each term in Eq.(17) can be written in the following form:

$$\tilde{c}_K e^{2 \sum_{k \in K} (p_k t + q_k)},$$

where $K$ is any multiset of the elements from $\{1, \ldots, m\}$ in which each element may appear at most twice and $\tilde{c}_K$ is a constant.

By observing Eq.(17), we see that the coefficient in front of this term is $4\beta c_1 - 2 \sum_{l \neq 1} c_l = 0$.

Combining the above arguments, we can conclude that $\sum_{l=1}^{m} c_l = 0$ and $4\beta c_1 - 2 \sum_{l \neq 1} c_l = 0$, thereby $(4\beta + 2)c_1 = 0$, i.e., $c_1 = 0$ because of $\beta \neq -\frac{1}{2}$. Following the same reasoning, we can similarly obtain $c_2 = 0, \ldots, c_m = 0$. Thus, the lemma is proved.

$\square$

## C PROOF OF PROPOSITION 1

*Proof.* We provide a detailed proof in the case of the gradient flow.

$\theta(t)$ is the solution to the gradient flow. Thus, the loss function is monotonically decreasing along the trajectory $\theta(t)$. So there exists a constant $C$ such that

$$\|\theta(t)\|_2 \leq C, \quad \forall t > 0.$$

because the loss function $\mathcal{J}$ is coercive.

Because $\theta(t)$ is uniformly bounded, there exists a subsequence $t_n \to \infty$ such that $\theta(t_n) \to \theta^*$, which is a critical point of $\mathcal{J}$.

We now have

$$\frac{d}{dt} \left( \mathcal{J}(\theta(t)) - \mathcal{J}(\theta^*) \right) = -\|\nabla \mathcal{J}(\theta(t))\|^2.$$

We first consider the case when $\epsilon \in (0, \frac{1}{2})$. Since $\mathcal{J}(\theta(t))$ is nonincreasing, we have $z(t) := \mathcal{J}(\theta(t)) - \mathcal{J}(\theta^*) \geq 0$, and as a consequence of Theorem 2,

$$z'(t) \leq -(z(t))^{2(1-\epsilon)} \implies \mathcal{J}(\theta(t)) - \mathcal{J}(\theta^*) = z(t) \leq K_1 t^{-\frac{1}{1-2\epsilon}}. \qquad (18)$$

Now since

$$\|\theta'(t)\|^2 = -z'(t),$$

we have

$$\int_t^{2t} \|\theta'(s)\|^2 ds = z(t) - z(2t) \leq z(t) \leq K_1 t^{-\frac{1}{1-2\epsilon}}.$$

Then by the Cauchy–Schwarz inequality,

$$\int_t^{2t} \|\theta'(s)\| ds \leq K_1 t^{-\frac{\epsilon}{1-2\epsilon}}.$$

Indeed it implies for all $t < t_n$,

$$\|\theta(t) - \theta(t_n)\|_2 = \|\int_t^T \theta'(s) ds\| \leq \int_t^T \|\theta'(s)\| ds \leq K_1 \sum_0^\infty (2^k t)^{-\frac{\epsilon}{1-2\epsilon}}$$

$$= K_1 \sum_0^\infty 2^{-\frac{\epsilon}{1-2\epsilon} k} t^{-\frac{\epsilon}{1-2\epsilon}} = K_2 t^{-\frac{\epsilon}{1-2\epsilon}}.$$

Letting $n$ tend to infinity in the above expression completes the proof.

If $\alpha = \frac{1}{2}$, we rewrite Eq.(18) as

$$\mathcal{J}(\theta(t)) - \mathcal{J}(\theta^*) \leq K_1 \exp(-t).$$

The rest of the proof is basically the same.

It is worth noting that implicit gradient descent (IGD) can be viewed as the backward Euler discretization of the gradient flow. Several works have studied the convergence properties of the backward Euler scheme. Applying Theorem 2.4 and Proposition 2.5 from Merlet & Pierre (2010), we can establish the desired result. □

## D  EXTENSION TO SECTION 4.2

### D.1  PROOF OF PROPOSITION 2

We now present the proof for the Proposition 2 in Section 4.2. Essentially, we need to process the functions so that it depends on a single variable, and then we can apply Lemma 1, which has already been proven.

*Proof.* Let $\{(w_k, b_k)\}_{k=1}^m$ be the admissible inner-layer parameters of the neural network, where $w_k \in \mathbb{R}^d$ and $b_k \in \mathbb{R}$.

Take $c_1, \ldots, c_m$ such that $\sum_{i=1}^m c_i \left(\alpha \tanh(w_i^\top x + b_i) + \beta w_{i,d} \tanh'(w_i^\top x + b_i)\right) = 0$ in $L^2(\Gamma)$.

Because of continuity, we obtain that $\sum_{i=1}^m c_i \left(\alpha \tanh(w_i^\top x + b_i) + \beta w_{i,d} \tanh'(w_i^\top x + b_i)\right) \equiv 0$ on $\Gamma$.

We denote the first $d - 1$ components of the vector $x \in \mathbb{R}^d$ by a new $(d-1)$-dimensional vector $\tilde{x}$, i.e., $x = (\tilde{x}^\top, x_d)^\top$. Using Assumption 1, we can rewrite the above equality as

$$\sum_{i=1}^m c_i \left(\alpha \tanh(\tilde{w}_i^\top \tilde{x} + \gamma w_{i,d} + b_i) + \beta w_{i,d} \tanh'(\tilde{w}_i^\top \tilde{x} + \gamma w_{i,d} + b_i)\right) = 0, \ \forall x = (\tilde{x}^\top, \gamma)^\top \in \Gamma.$$

Note that $\lambda_{d-1}(\Gamma) > 0$, so there exists an open ball $B$ contained in $\mathbb{R}^{d-1}$, such that

$$\sum_{i=1}^m c_i \left(\alpha \tanh(\tilde{w}_i^\top x + \gamma w_{i,d} + b_i) + \beta w_{i,d} \tanh'(\tilde{w}_i^\top x + \gamma w_{i,d} + b_i)\right) = 0, \quad \forall \tilde{x} \in B.$$

which is equivalent to

$$\sum_{i=1}^m c_i \left(\alpha \tanh(\tilde{w}_i^\top x + \gamma w_{i,d} + b_i) + \beta w_{i,d} \tanh'(\tilde{w}_i^\top x + \gamma w_{i,d} + b_i)\right) = 0, \quad \forall \tilde{x} \in \mathbb{R}^{d-1}. \quad (19)$$

because $\tanh$ is an analytic function.

For any $1 \leq k < j \leq m$, define $A_{k,j}, B_{k,j}$ as follows:

$$A_{k,j} = \{\tilde{x} \in \mathbb{R}^{d-1} : (\tilde{w}_k - \tilde{w}_j)^\top \tilde{x} = 0\}, \quad B_{k,j} = \{\tilde{x} \in \mathbb{R}^{d-1} : (\tilde{w}_k + \tilde{w}_j)^\top \tilde{x} = 0\}.$$

The sets $A_{k,j}, B_{k,j}$ are subspaces of dimension $d - 2$, so $\bigcup_{1 \leq k < j \leq m}(A_{k,j} \cup B_{k,j})$ has $\lambda_{d-1}$ -measure zero. This implies that we can choose some $e \in \mathbb{R}^{d-1}$ with $\|e\|_2 = 1$ such that for all $1 \leq k < j \leq m$,

$$p_k := \tilde{w}_k^\top e \neq \pm \tilde{w}_j^\top e =: p_j.$$

By Eq.(19), we have for $\varepsilon \in \mathbb{R}$ and $\tilde{x} = \varepsilon e$,

$$\sum_{i=1}^m c_i \left(\alpha \tanh(\tilde{w}_i^\top e \varepsilon + \gamma w_{i,d} + b_i) + \beta w_{i,d} \tanh'(\tilde{w}_i^\top e \varepsilon + \gamma w_{i,d} + b_i)\right) = 0, \quad \forall \tilde{x} \in \mathbb{R}^{d-1}.$$

Note that $p_k \neq \pm p_j$ for all $1 \leq k < j \leq m$, we can obtain that $c_k = 0$ for all $1 \leq k \leq m$ by using Lemma 1.

So we can conclude that the functions

$$\alpha \tanh(w_1^\top x + b_1) + \beta w_{1,d} \tanh'(w_1^\top x + b_1), \ldots, \alpha \tanh(w_m^\top x + b_m) + \beta w_{m,d} \tanh'(w_m^\top x + b_m)$$

are linearly independent in $L^2(\Gamma)$.

Recall that $u = \sum_{k=1}^m a_k \sigma(w_k^\top x + b_k)$. Under Assumption 1, without loss of generality, we can express the outward normal vector as $n = (0, \ldots, 0, 1)$ on the flat segment $\Gamma$. Then we can rewrite the outward derivative $\frac{\partial u}{\partial n}$ as

$$\frac{\partial u}{\partial n} = \sum_{k=1}^m a_k w_{k,d} \sigma'(w_k^\top x + b_k).$$

Define Gram matrix $G = (G_{ij})_{1 \leq i,j \leq m}$, where

$$G_{ij} = \int_\Gamma \left( \alpha \tanh(w_i^\top x + b_i) + \beta w_{i,d} \tanh'(w_i^\top x + b_i) \right)$$
$$\left( \alpha \tanh(w_j^\top x + b_j) + \beta w_{j,d} \tanh'(w_j^\top x + b_j) \right) \mathrm{d}x.$$

The Gram matrix $G$ is positive definite by the linear independence, and we denote the smallest eigenvalue of $G$ as $\lambda_{\min} > 0$.

Then we have

$$\left\| \alpha u + \beta \frac{\partial u}{\partial n} \right\|_{L^2(\Gamma)}^2 = \left\| \sum_{k=1}^m a_k \left( \alpha \tanh(w_k^\top x + b_k) + \beta w_{k,d} \tanh'(w_k^\top x + b_k) \right) \right\|_{L^2(\Gamma)}^2 \tag{20}$$
$$= a^\top G a \geq \lambda_{\min} \|a\|_2^2.$$

$\square$

## D.2 DISCUSSION ON EMPIRICAL LOSS FUNCTION

In practice, we approximate the loss function equation 8 using discrete sample points. Let $X^{(1)} = \{x_k^{(1)}\}_{k=1}^{n_1} \subset \Omega$ (interior points) and $X^{(2)} = \{x_k^{(2)}\}_{k=1}^{n_2} \subset \partial\Omega$ (boundary points). Under Dirichlet boundary condition, the empirical loss is written as:

$$\mathcal{J}_{\mathrm{emp}}(a) = \frac{1}{n_1} \sum_{k=1}^{n_1} \left( \mathcal{L}u_\theta(x_k^{(1)}) - f(x_k^{(1)}) \right)^2 + \frac{\lambda}{n_2} \sum_{k=1}^{n_2} \left( u_\theta(x_k^{(2)}) - g(x_k^{(2)}) \right)^2. \tag{21}$$

Under Assumption 1, we assume $n_2 \geq m$ and consider a subset $\tilde{X}^{(2)} = \{x_k^{(2)}\}_{k=1}^m \subset \Gamma$. For each $1 \leq k \leq m$, define the activation vector,

$$\sigma_k(\tilde{X}^{(2)}) = \left( \tanh(w_k^\top x_1^{(2)} + b_k), \ldots, \tanh(w_k^\top x_m^{(2)} + b_k) \right)^\top.$$

Then we can establish the following result, which is a discrete version of Proposition 2. However, the techniques needed are not the same.

**Proposition 4** (Linear independence on discrete points). *For any $m$ vectors $\{\tilde{w}_k\}_{k=1}^m \subset \mathbb{R}^{d-1}$ with $\tilde{w}_i \neq \pm\tilde{w}_j$ ($i \neq j$), and for sufficiently small $w_{k,d}, b_k \in \mathbb{R}$, the vectors $\{\sigma_k(\tilde{X}^{(2)})\}_{k=1}^m$ are linearly independent for almost all $\tilde{X}^{(2)} \in \Gamma^m$.*

Here, "almost all" means the condition holds generically, making it practically feasible to find suitable sample points. This leads to our main convergence guarantee for the empirical loss.

**Theorem 5** (Global convergence of empirical loss). *Under Assumption 1, we consider the empirical loss $\mathcal{J}_{emp}(a)$ in equation 21 with randomly sampled point sets $X^{(1)}, X^{(2)}$. When inner-layer parameters $\{(w_k, b_k)\}_{k=1}^m$ are initialized as: $\tilde{w}_i \sim \mathcal{N}(0, I_{d-1})$ i.i.d ; $|w_{i,d}|, |b_i| < \delta$ (sufficiently small) then with probability 1, all convergence results of Proposition 1 hold for both gradient flow equation 9 and implicit gradient descent equation 10.*

We prove the above proposition and theorem in Section D.2.1.

### D.2.1 PROOF OF PROPOSITION 4 AND THEOREM 5

The proof of Proposition 4 requires special treatment for the selection of boundary sampling points. We first prove the following lemma, which shows that there exist suitable sampling points to ensure linear independence.

**Lemma 2.** *For any choice of $m$ vectors $\tilde{w}_1, \ldots, \tilde{w}_m \in \mathbb{R}^{d-1}$ such that $\tilde{w}_i \neq \pm\tilde{w}_i$ if $i \neq j$, and for sufficiently small scalars $w_{1,d}, \ldots, w_{m,d} \in \mathbb{R}$ and $b_1, \ldots, b_m \in \mathbb{R}$, there exists a set $X = \{x_k\}_{k=1}^m \subset \Gamma$ such that the vectors $\sigma_1(X), \ldots, \sigma_m(X)$ are linearly independent, where $w_r = (\tilde{w}_r^\mathsf{T}, w_{r,d})^\mathsf{T}$.*

*Proof.* Let $v_i = (\tanh(w_1^\mathsf{T} x_i + b_1), \ldots, \tanh(w_m^\mathsf{T} x_i + b_m))^\mathsf{T}$. We want to seeek $\{x_i\}_{i=1}^m$ such that $v_1, \ldots, v_m$ are linear independent. We use induction to sequentially find appropriate $x_1, x_2, \ldots, x_m$.

First, we can choose $x_1$ arbitrarily in $\Gamma$ such that $w_1^\mathsf{T} x_1 + b_1 \neq 0$. Because $\tanh(w_1^\mathsf{T} x_1 + b_1) \neq 0$, $v_1$ is linear independent.

Next, we assume that $x_1, \ldots, x_{k-1}$ have been chosen such that $v_1, \ldots, v_{k-1}$ are linearly independent. We need to choose $x_k$ such that $v_1, \ldots, v_k$ are linearly independent.

Choose $e \in \mathbb{R}^{d-1}$ with $\|e\|_2 = 1$ such that $p_k := \tilde{w}_k^\mathsf{T} e \neq \pm\tilde{w}_j^\mathsf{T} e =: p_j$ for all $1 \leq k < j \leq m$.

Note that $\Gamma = \partial\Omega \cap B(x_0, r)$, we take $x_k = ((\tilde{x}_0 + \varepsilon e)^\mathsf{T}, \beta)^\mathsf{T}$ which is naturally in $\Gamma$ for sufficiently small $\varepsilon$.

Take any non-zero vector $b \in \mathbb{R}^m$ such that $b$ is orthogonal to $v_1, \ldots, v_{k-1}$, i.e., $b^\mathsf{T} v_i = 0$ for all $1 \leq i \leq k - 1$.

Consider the function $F(\varepsilon) = \sum_{i=1}^m b_i \tanh(p_i\varepsilon + w_{i,d}\gamma + b_i)$. Because $b \neq 0$, $F(\varepsilon)$ is not constant zero by Lemma 1.

Take any $\varepsilon_0$ such that $F(\varepsilon_0) \neq 0$ and $x_k = ((\tilde{x}_0 + \varepsilon e)^\mathsf{T}, \gamma)^\mathsf{T}$, then we can obtain $v_k \notin \mathrm{span}\{v_j : 1 \leq j \leq k - 1\}$. Otherwise, $b$ is orthogonal to $v_k$ and then $F(\varepsilon_0) = 0$.

So by induction we can obtain a set $X = \{x_i\}_{i=1}^m \subset \Gamma$ such that $v_1, \ldots, v_m$ are linear independent, which is equivalent to $\sigma_1(X), \ldots, \sigma_m(X)$ are linear independent.

$\square$

The proof of the Proposition 4 is given below. That is, based on the existence, we further show that such sampling points are almost everywhere.

*Proof.* Define matrix $\sigma(X) = (\sigma_1(X), \ldots, \sigma_m(X))$. We want to show that for almost all $X = (x_1, \ldots, x_n) \in \Gamma^n$, $\sigma(X)$ is full-rank.

By Lemma 2, we can find a set $X^* = \{x_k^*\}_{k=1}^m \subset \Gamma$ such that $\sigma_1(X^*), \ldots, \sigma_m(X^*)$ are linear independent, i.e. $\det(\sigma(X^*)) \neq 0$. Note that $\det\sigma(\cdot)$ is an analytic function defined on $\Gamma^m$, so its zero set is of zero measure. This means that for almost all $X = (x_1, \ldots, x_m) \in \Gamma^m$, $\sigma(X)$ is full-rank.

Then for any $n \geq m$, for almost all $X = (x_1, \ldots, x_n) \in \Gamma^n$, $\sigma(X)$ is full-rank. $\square$

Below, based on the previous conclusions, we present the proof of Theorem 5.

*Proof.* We only prove the second conclusion.

Because we can, with probability 1, select sampling points $\tilde{X}^{(2)}$ and internal parameters that satisfy the conditions of the above lemma. By linear independence, the Gram matrix $G = (G_{ij})_{i,j=1}^m$ is positive definite, we $G_{ij} = \sigma_i(\tilde{X}^{(2)})^\mathsf{T}\sigma_j(\tilde{X}^{(2)})$.

So $\mathcal{J}_{\mathrm{emp}}(a) \geq \frac{\lambda}{n_2} a^\mathsf{T} G a \geq \frac{\lambda}{n_2}\lambda_{\min}(G)\|a\|_2^2$, which implies $J$ is coercive. And then we can apply Proposition 1. $\square$

# E  PROOF OF PROPOSITION 3

*Proof.* We take

$$\mathcal{L}u = -\mathrm{div}(|\nabla u|^{p-2}\nabla u) + q(x)u + h(u), \text{ where } p \geq 2, \ q(x) \geq 0, \ h(u)u \geq 0 \tag{22}$$

as an example for the proof.

Note that in $L^2(\Omega)$ inner-product space, we have

$$\begin{aligned}
\langle \mathcal{L}(\tilde{u}), \tilde{u} \rangle =& \langle -\mathrm{div}(|\nabla \tilde{u}|^{p-2}\nabla \tilde{u}) + q(x)\tilde{u} + h(\tilde{u}), \tilde{u} \rangle \\
=& \int_\Omega \|\tilde{u}\|^p + q(x)\tilde{u}^2 + h(\tilde{u})\tilde{u} \, \mathrm{d}x \text{ (by integration by parts)} \\
\geq& \int_\Omega \|\tilde{u}\|^p \, \mathrm{d}x \text{ (by the assumption of } q(x) \text{ and } h(u)) \\
\geq& C\|\tilde{u}\|^2_{L^2(\Omega)} \text{ (by the Sobolev inequality)}
\end{aligned}$$

where $C$ is a constant depending on the domain $\Omega$ and the Sobolev embedding constant.

On the other hand, using the Cauchy–Schwarz inequality, we have

$$\langle \mathcal{L}(\tilde{u}), \tilde{u} \rangle \leq \|\mathcal{L}(\tilde{u})\|_{L^2(\Omega)}\|\tilde{u}\|_{L^2(\Omega)}.$$

So we can conclude that there exists a constant $C > 0$ such that

$$\|u\|_{L^2(U)} \leq \|\tilde{u}\|_{L^2(U)} \leq \|\tilde{u}\|_{L^2(\Omega)} \leq C\|\mathcal{L}(\tilde{u})\|_{L^2(\Omega)} \leq C(\|\mathcal{L}(\tilde{u}) - f\|_{L^2(\Omega)} + \|f\|_{L^2(\Omega)})$$

$$\square$$

# F  DEEP RITZ-TYPE INTERIOR CONTROL

The Deep Ritz method employs the energy functional $\mathcal{E}(u_\theta)$ as its interior loss. We demonstrate coercivity through two canonical examples.

**Example 1: $p$-Laplace equation**  The $p$-Laplace equation

$$-\mathrm{div}(|\nabla u|^{p-2}\nabla u) = f(x) \tag{23}$$

generalizes the classical Laplace equation ($p = 2$) to model nonlinear diffusion processes. It arises in non-Newtonian fluid dynamics ($1 < p < 2$ for shear-thinning fluids) and image processing (edge-preserving denoising). The associated energy functional

$$\mathcal{E}(u) = \int_\Omega \frac{1}{p}|\nabla u|^p - f(x)u \, \mathrm{d}x \tag{24}$$

exhibits $p$-growth conditions, making its analysis distinct from quadratic elliptic problems. A fundamental result of the variational theory: the energy functional $Eq.(24)$ is coercive, i.e., there exist constants $c, C$,

$$\mathcal{E}(u) \geq c\|u\|^p_{H_0^{1,p}} - C.$$

**Example 2: stationary Allen–Cahn equation**  This phase-field model

$$-\epsilon^2\Delta u + (u^3 - u) = 0 \tag{25}$$

describes phase separation in binary alloys, with $\epsilon$ controlling interface width. Its double-well potential energy

$$\mathcal{E}(u) = \int_\Omega \frac{\epsilon^2}{2}|\nabla u|^2 + \frac{1}{4}(u^2 - 1)^2 \, \mathrm{d}x \tag{26}$$

forces solutions toward $\pm 1$ (pure phases) with transition zones of $O(\epsilon)$ width. Also, the energy functional $Eq.(26)$ is coercive, i.e., there exist constants $c, C$,

$$\mathcal{E}(u) \geq c\|u\|^2_{H^1} - C.$$

Therefore, we can use the technique from Section 4.3 to prove that the loss function $\mathcal{J}$ is coercive with respect to $a$, thereby establishing the convergence of Deep Ritz method.

## G    PROOF OF THEOREM 1

Here we provide the proof of Theorem 1, with the main idea inspired by previous works Gao et al. (2023); Xu et al. (2024a). However, since we deal with more general linear operators, some calculations require greater care compared to the procedures in previous works. Although the proof strategy is clear, the details are quite involved. We first give a brief outline of the approach, and then rigorously justify each step through a series of lemmas.

Let us first review some notations from the main text and introduce several new ones. We focus on the linear PDE with the following form:

$$
\begin{cases}
\sum_{k=0}^{+\infty} \sum_{|\xi|=k} c_\xi(x) \partial^\xi u = f, & x \in \Omega, \\
\alpha u + \beta \frac{\partial u}{\partial n} = g, & x \in \partial\Omega,
\end{cases}
\tag{27}
$$

where the linear operator $\mathcal{L}$ satisfies Definition 1, and $f, g$ are bounded continuous functions. In the following, we assume that $\|x\|_2 \leq \frac{\sqrt{3}}{2}$ for $x \in \overline{\Omega}$.

We consider a two-layer neural network of the following form,

$$
u_\theta(x) = \frac{1}{\sqrt{m}} \sum_{k=1}^{m} a_k \tanh(w_k^\mathsf{T} x + b_k).
$$

To handle the bias term more conveniently, we consider augmenting both $x$ and the PDE. We define $y = (x^\mathsf{T}, \frac{1}{2})^\mathsf{T}$ for $x \in \Omega$, then we have $\|y\|_2 \leq 1$. For Eq.(27), we will rewrite the equation about $y$, and for simplicity, we still use the same notation:

$$
\begin{cases}
\sum_{k=0}^{+\infty} \sum_{|\xi|=k} c_\xi(y) \partial^\xi u = f, & y \in \Omega \times \{\frac{1}{2}\}, \\
\alpha u + \beta \frac{\partial u}{\partial n} = g, & y \in \partial\Omega \times \{\frac{1}{2}\},
\end{cases}
\tag{28}
$$

where the original $d$-dimensional multi-index $\xi$ is augmented to $(d+1)$-dimensional multi-index , which is still denoted as $\xi = (\xi, 0)$. And we rewrite the neural network as

$$
u_\theta(x) = \frac{1}{\sqrt{m}} \sum_{k=1}^{m} a_k \tanh(w_k^\mathsf{T} y),
\tag{29}
$$

where $a_k \in \mathbb{R}$ and $w_k \in \mathbb{R}^{d+1}$ for $\leq k \leq m$.

In the framework of PINNs, we focus on the empirical risk minimization problem. Given training samples $\{y_p^{(1)}\}_{p=1}^{n_1} \subset \Omega \times \{\frac{1}{2}\}$ and $\{y_p^{(2)}\}_{p=1}^{n_2} \subset \partial\Omega \times \{\frac{1}{2}\}$, we aim to minimize the empirical loss function as follows,

$$
\begin{aligned}
\mathcal{J}_{\text{emp}}(\theta) = {} & \frac{1}{n_1} \sum_{i=1}^{n_1} \frac{1}{2} \left| \sum_{k=0}^{+\infty} \sum_{|\xi|=k} c_\xi(y) \partial^\xi u_\theta\left(y_i^{(1)}\right) - f\left(y_i^{(1)}\right) \right|^2 \\
& + \frac{\lambda}{n_2} \sum_{j=1}^{n_2} \frac{1}{2} \left| \alpha u_\theta\left(y_j^{(2)}\right) + \beta \frac{\partial u_\theta\left(y_j^{(2)}\right)}{\partial n} - g\left(y_j^{(2)}\right) \right|^2 .
\end{aligned}
\tag{30}
$$

where $\theta = \{(a_k, w_k)\}_{k=1}^{m} \in \mathbb{R}^{m(d+2)}$ are all trainable parameters in Eq.(29).

We consider the gradient flow training dynamics: for $1 \leq k \leq m$

$$
\frac{\mathrm{d}w_k(t)}{\mathrm{d}t} = -\frac{\partial \mathcal{J}_{\text{emp}}(\theta(t))}{\partial w_k}, \quad \frac{\mathrm{d}a_k(t)}{\mathrm{d}t} = -\frac{\partial \mathcal{J}_{\text{emp}}(\theta(t))}{\partial a_k}.
\tag{31}
$$

Let

$$
s_p(\theta) = \frac{1}{\sqrt{n_1}} \left( \sum_{k=0}^{+\infty} \sum_{|\xi|=k} c_\xi(y) \partial^\xi u_\theta\left(y_p^{(1)}\right) - f\left(y_p^{(1)}\right) \right), \forall 1 \leq p \leq n_1,
$$

and

$$h_j(\theta) = \sqrt{\frac{\lambda}{n_2}} \left( \alpha u_\theta \left( y_j^{(2)} \right) + \beta \frac{\partial u_\theta \left( y_j^{(2)} \right)}{\partial n} - g \left( y_j^{(2)} \right) \right), \forall 1 \leq j \leq n_2.$$

Then we have

$$\mathcal{J}_{\text{emp}}(\theta) = \frac{1}{2} (\|s(\theta)\|_2^2 + \|h(\theta)\|_2^2),$$

where vectors $s(\theta) = (s_1(\theta), \ldots, s_{n_1}(\theta))^\mathsf{T}$ and $h(\theta) = (h_1(\theta), \ldots, h_{n_2}(\theta))^\mathsf{T}$. Therefore, for $1 \leq k \leq m$,

$$\frac{\mathrm{d} w_k}{\mathrm{d} t} = -\frac{\partial \mathcal{J}_{\text{emp}}(\theta)}{\partial w_k}$$

$$= -\sum_{p=1}^{n_1} s_p(\theta) \cdot \frac{\partial s_p(\theta)}{\partial w_k} - \sum_{k=1}^{n_2} h_k(\theta) \cdot \frac{\partial h_k(\theta)}{\partial w_k},$$

and

$$\frac{\mathrm{d} a_k}{\mathrm{d} t} = -\frac{\partial \mathcal{J}_{\text{emp}}(\theta)}{\partial a_k}$$

$$= -\sum_{p=1}^{n_1} s_p(\theta) \cdot \frac{\partial s_p(\theta)}{\partial a_k} - \sum_{k=1}^{n_2} h_k(\theta) \cdot \frac{\partial h_k(\theta)}{\partial a_k}.$$

Using the chain rule, after simple computation, we can derive the following dynamics:

$$\frac{\mathrm{d}}{\mathrm{d} t} \begin{bmatrix} s(\theta) \\ h(\theta) \end{bmatrix} = - \left( \mathbf{G}(\theta) + \widetilde{\mathbf{G}}(\theta) \right) \begin{bmatrix} s(\theta) \\ h(\theta) \end{bmatrix}, \tag{32}$$

where $\mathbf{G}(\theta)$ and $\widetilde{\mathbf{G}}(\theta)$ are the Gram matrices for the dynamics, defined as

$$\mathbf{G}(\theta) = \mathbf{D}^\top \mathbf{D}, \quad \mathbf{D} = \begin{bmatrix} \frac{\partial s_1}{\partial \mathbf{W}} & \cdots & \frac{\partial s_{n_1}}{\partial \mathbf{W}} & \frac{\partial h_1}{\partial \mathbf{W}} & \cdots & \frac{\partial h_{n_2}}{\partial \mathbf{W}} \end{bmatrix}, \tag{33}$$

where $\mathbf{W} = (w_1^\mathsf{T}, \ldots, w_m^\mathsf{T})^\mathsf{T}$, and

$$\widetilde{\mathbf{G}}(\theta) = \widetilde{\mathbf{D}}^\top \widetilde{\mathbf{D}}, \quad \widetilde{\mathbf{D}} = \begin{bmatrix} \frac{\partial s_1}{\partial \mathbf{a}} & \cdots & \frac{\partial s_{n_1}}{\partial \mathbf{a}} & \frac{\partial h_1}{\partial \mathbf{a}} & \cdots & \frac{\partial h_{n_2}}{\partial \mathbf{a}} \end{bmatrix}, \tag{34}$$

where $\mathbf{a} = (a_1, \ldots, a_m)^\mathsf{T}$. Moreover, we rewrite $\theta = (\mathbf{W}, \mathbf{a})$ and define

$$\mathbf{G}^\infty = \mathbb{E}_{\mathbf{W} \sim \mathcal{N}(0, \mathbf{I}), \mathbf{a} \sim \text{Unif}(\{-1, 1\}^m)} \mathbf{G}(\mathbf{W}, \mathbf{a})$$

and

$$\widetilde{\mathbf{G}}^\infty = \mathbb{E}_{\mathbf{W} \sim \mathcal{N}(0, \mathbf{I}), \mathbf{a} \sim \text{Unif}(\{-1, 1\}^m)} \widetilde{\mathbf{G}}(\mathbf{W}, \mathbf{a}).$$

Now that we have established all the basic definitions, we will first outline our proof strategy.

**Proof sketch:**

(i) To prove that the expectation of the Gram matrices $\mathbf{G}^\infty, \widetilde{\mathbf{G}}^\infty$ are positive definite (Lemma 3).

(ii) To show that, with high probability, the Gram matrix at initialization $\mathbf{G}(\mathbf{W}(0), \mathbf{a}(0))$, $\widetilde{\mathbf{G}}(\mathbf{W}(0), \mathbf{a}(0))$ are close to $\mathbf{G}^\infty, \widetilde{\mathbf{G}}^\infty$ respectively, thereby implying that the Gram matrix $\mathbf{G}(\mathbf{W}(0), \mathbf{a}(0)), \widetilde{\mathbf{G}}(\mathbf{W}(0), \mathbf{a}(0))$ are positive definite with high probability (Lemma 4).

(iii) To prove that the Gram matrix $\mathbf{G}(\mathbf{W}, \mathbf{a}), \widetilde{\mathbf{G}}(\mathbf{W}, \mathbf{a})$ are stable with respect to $\mathbf{W}$ and $\mathbf{a}$, that is, if the parameters are perturbed slightly, the corresponding Gram matrix remains close to the original (Lemma 5).

(iv) To prove that, during the evolution by gradient flow Eq.(31), the parameters do not change much. Combining this with the previous three results, we know that the Gram matrix $\mathbf{G}(\mathbf{W}(t), \mathbf{a}(t)), \widetilde{\mathbf{G}}(\mathbf{W}(t), \mathbf{a}(t))$ remain positive definite with high probability throughout the evolution, and we can estimate its minimal eigenvalue. This allows us to prove that the loss decreases at a certain rate.

Next, we will carry out the above proof strategy step by step through a series of lemmas.

**Lemma 3** (Positive definiteness of $\mathbf{G}^\infty, \widetilde{\mathbf{G}}^\infty$). *The expectation of the Gram matrices $\mathbf{G}^\infty, \widetilde{\mathbf{G}}^\infty$ are positive definite.*

*Proof.* **Part 1:** we first prove the positive definiteness of $\mathbf{G}^\infty$.

We denote $\varphi(y; w) = \sum_{k=0}^{+\infty} \sum_{|\xi|=k} c_\xi(y) \tanh^{(|\xi|)}(w^\mathsf{T} y) w^\xi$, where $w \in \mathbb{R}^{d+1}$. Then,

$$\frac{\partial s_p}{\partial w_k} = \frac{1}{\sqrt{n_1}} \frac{a_k}{\sqrt{m}} \frac{\partial \varphi(y_p^{(1)}; w_k)}{\partial w}.$$

Similarly, let $\psi(y; w) = \alpha \tanh(w^T y) + \beta \tanh'(w^\mathsf{T} y) w^\mathsf{T} n(y)$, where $n(y)$ is the outer normal direction on the point $y \in \partial\Omega \times \{\frac{1}{2}\}$. Then

$$\frac{\partial h_j}{\partial w_k} = \frac{1}{\sqrt{n_2}} \frac{a_k}{\sqrt{m}} \frac{\partial \psi(y_j^{(2)}; w_k)}{\partial w}.$$

With these notations, we deduce that

$$G_{p,j}^\infty = \begin{cases} \frac{1}{n_1} \mathbb{E}_{w \sim \mathcal{N}(0,\mathbf{I})} \left\langle \frac{\partial \varphi(y_p^{(1)}; w)}{\partial w}, \frac{\partial \varphi(y_j^{(1)}; w)}{\partial w} \right\rangle, & 1 \le p \le n_1, \, 1 \le j \le n_1, \\ \frac{1}{\sqrt{n_1 n_2}} \mathbb{E}_{w \sim \mathcal{N}(0,I)} \left\langle \frac{\partial \varphi(y_p^{(1)}; w)}{\partial w}, \frac{\partial \psi(y_j^{(2)}; w)}{\partial w} \right\rangle, & 1 \le p \le n_1, \, n_1 + 1 \le j \le n_1 + n_2, \\ \frac{1}{n_2} \mathbb{E}_{w \sim \mathcal{N}(0,I)} \left\langle \frac{\partial \psi(y_p^{(2)}; w)}{\partial w}, \frac{\partial \psi(y_p^{(2)}; w)}{\partial w} \right\rangle, & n_1 + 1 \le p, j \le n_1 + n_2, \end{cases}$$

where $G_{p,j}^\infty$ denotes the $(p, j)$-th entry of $\mathbf{G}^\infty$.

To prove this lemma, we need tools from functional analysis. Let $\mathcal{H}$ be a Hilbert space of integrable $(d + 1)$-dimensional vector fields on $\mathbb{R}^{d+1}$, i.e., $f \in \mathcal{H}$ if $\mathbb{E}_{w \sim \mathcal{N}(0,I)}[\|f(w)\|_2^2] < \infty$. The inner product for any two elements $f, g \in \mathcal{H}$ is $\mathbb{E}_{w \sim \mathcal{N}(0,\mathbf{I})}[\langle f(w), g(w) \rangle]$. Thus, to show that $\mathbf{G}^\infty$ is strictly positive definite, it suffices to demonstrate that

$$\frac{\partial \varphi(y_1^{(1)}; w)}{\partial w}, \dots, \frac{\partial \varphi(y_{n_1}^{(1)}; w)}{\partial w}, \frac{\partial \psi(y_1^{(2)}; w)}{\partial w}, \dots, \frac{\partial \psi(y_{n_2}^{(2)}; w)}{\partial w} \in \mathcal{H}$$

are linearly independent. Suppose there exist coefficients $c_1^{(1)}, \dots, c_{n_1}^{(1)}, c_1^{(2)}, \dots, c_{n_2}^{(2)} \in \mathbb{R}$ such that

$$c_1^{(1)} \frac{\partial \varphi(y_1^{(1)}; w)}{\partial w} + \cdots + c_{n_1}^{(1)} \frac{\partial \varphi(y_{n_1}^{(1)}; w)}{\partial w} + c_1^{(2)} \frac{\partial \psi(y_1^{(2)}; w)}{\partial w} + \cdots + c_{n_2}^{(2)} \frac{\partial \psi(y_{n_2}^{(2)}; w)}{\partial w} = 0 \text{ in } \mathcal{H}.$$

This implies that

$$c_1^{(1)} \frac{\partial \varphi(y_1^{(1)}; w)}{\partial w} + \cdots + c_{n_1}^{(1)} \frac{\partial \varphi(y_{n_1}^{(1)}; w)}{\partial w} + c_1^{(2)} \frac{\partial \psi(y_1^{(2)}; w)}{\partial w} + \cdots + c_{n_2}^{(2)} \frac{\partial \psi(y_{n_2}^{(2)}; w)}{\partial w} = 0 \quad (35)$$

for all $w \in \mathbb{R}^{d+1}$.

We first compute the derivatives of $\varphi$ and $\psi$. Differentiating $\psi(y; w)$ $l$ times with respect to $w$, we have

$$\frac{\partial^l \psi(y; w)}{\partial w^l} = \alpha \tanh^{(l)}(w^T y) y^{\otimes(l)} + \beta \sum_{s=0}^l \tanh^{(l-s+1)}(w^\mathsf{T} y) y^{\otimes(l-s)} \otimes \frac{\partial^s w^\mathsf{T} n(y)}{\partial w^s},$$

where $\otimes$ denotes the tensor product.

Differentiating $\varphi(y; w)$ $l$ times with respect to $w$, similar to the Leibniz rule for the $l$-th derivative of the product of two scalar functions, we obtain

$$\frac{\partial^l \varphi(y; w)}{\partial w^l} = \sum_k \sum_{|\xi|=k} c_\xi(y) \sum_{s=0}^l \tanh^{(l-s+|\xi|)}(w^\mathsf{T} y) y^{\otimes(l-s)} \otimes \frac{\partial^s w^\alpha}{\partial w^s}.$$

Note that, the set

$$y_1^{(1),\otimes(n_1+n_2-1)}, \ldots, y_{n_1}^{(1),\otimes(n_1+n_2-1)}, y_1^{(2),\otimes(n_1+n_2-1)}, \ldots, y_{n_2}^{(2),\otimes(n_1+n_2-1)}$$

is independent (see Lemma G.6 in Du et al. (2019a)).

This observation motivates us to differentiate both sides of Eq.(35) exactly $l-1 = n_1+n_2-1+d|\tilde{\xi}|$ times for $w$, where $\tilde{\xi}$ is defined in Definition 1. Thus, we have

$$c_1^{(1)} \frac{\partial^l \varphi(y_1^{(1)}; w)}{\partial w^l} + \cdots + c_{n_1}^{(1)} \frac{\partial^l \varphi(y_{n_1}^{(1)}; w)}{\partial w^l} + c_1^{(2)} \frac{\partial^l \psi(y_1^{(2)}; w)}{\partial w^l} + \cdots + c_{n_2}^{(2)} \frac{\partial^l \psi(y_{n_2}^{(2)}; w)}{\partial w^l} = 0.$$

By substituting the previous results into this equation, we have

$$\sum_{p=1}^{n_1} c_p^{(1)} \sum_k \sum_{|\xi|=k} c_\xi(y_p^{(1)}) \sum_{s=0}^{d|\tilde{\xi}|} \tanh^{(l-s+|\xi|)}(w^\mathsf{T} y_p^{(1)}) y_p^{(1),\otimes(l-s)} \otimes \frac{\partial^s w^\alpha}{\partial w^s} + \sum_{j=1}^{n_2} c_j^{(2)}$$

$$\left[ \alpha \tanh^{(l)}(w^T y_j^{(2)}) y_j^{(2),\otimes(l)} + \beta \sum_{s=0}^{d} \tanh^{(l-s+1)}(w^\mathsf{T} y_j^{(2)}) y_j^{(2),\otimes(l-s)} \otimes \frac{\partial^s w^\mathsf{T} n(y_j^{(2)})}{\partial w^s} \right] = 0,$$

where some higher-order derivative terms naturally vanish, so we have omitted them from the expression. Reorganizing the above equality as a linear combination in terms of

$$y_1^{(1),\otimes(n_1+n_2-1)}, \ldots, y_{n_1}^{(1),\otimes(n_1+n_2-1)}, y_1^{(2),\otimes(n_1+n_2-1)}, \ldots, y_{n_2}^{(2),\otimes(n_1+n_2-1)},$$

we explicitly list the coefficient in front of each term as follows:

$$\left[ \sum_{s=0}^{d|\tilde{\xi}|} \sum_k \sum_{|\xi|=k} c_\xi(y_p^{(1)}) \tanh^{(l-s+|\xi|)}(w^\mathsf{T} y_p^{(1)}) y_p^{(1),\otimes(l-s)} \otimes \frac{\partial^s w^\alpha}{\partial w^s} \right] c_p^{(1)} = 0, \ \forall 1 \le p \le n_1, \quad (36)$$

and for $1 \le j \le n_2$,

$$\left[ \alpha \tanh^{(l)}(w^T y_j^{(2)}) y_j^{(2),\otimes(l)} + \beta \sum_{s=0}^{d} \tanh^{(l-s+1)}(w^\mathsf{T} y_j^{(2)}) y_j^{(2),\otimes(l-s)} \otimes \frac{\partial^s w^\mathsf{T} n(y_j^{(2)})}{\partial w^s} \right] c_j^{(2)} = 0.$$
$$(37)$$

Note that under Definition 1, the term inside the braces $[]$ has a leading order. Therefore, as $w$ approaches infinity, the term in the brackets will not vanish. As a result, we can obtain

$$c_p^{(1)} = 0, \ c_j^{(2)} = 0, \quad \forall 1 \le p \le n_1, \ 1 \le j \le n_2.$$

So we can obtain that

$$\frac{\partial \varphi(y_1^{(1)}; w)}{\partial w}, \ldots, \frac{\partial \varphi(y_{n_1}^{(1)}; w)}{\partial w}, \frac{\partial \psi(y_1^{(2)}; w)}{\partial w}, \ldots, \frac{\partial \psi(y_{n_2}^{(2)}; w)}{\partial w} \in \mathcal{H}$$

are linearly independent. And $\mathbf{G}^\infty$ is positive definite.

**Part 2:** we prove that $\widetilde{\mathbf{G}}^\infty$ is positive definite. Note that

$$\frac{\partial s_p}{\partial a_k} = \frac{1}{\sqrt{n_1}} \frac{a_k}{\sqrt{m}} \varphi(y_p^{(1)}; w_k), \text{ and } \frac{\partial h_j}{\partial a_k} = \frac{1}{\sqrt{n_2}} \frac{a_k}{\sqrt{m}} \psi(y_j^{(2)}; w_k).$$

Therefore, the subsequent proof proceeds in the same way as before. $\square$

**Lemma 4.** *Define $\lambda_0, \tilde{\lambda}_0$ to be the minimal eigenvalue of $\mathbf{G}^\infty, \widetilde{\mathbf{G}}^\infty$ respectively. If $m = \Omega\left(\frac{d^{2|\tilde{\xi}|}}{\min\{\lambda_0^2, \tilde{\lambda}_0^2\}} \log\left(\frac{n_1+n_2}{\delta}\right)\right)$, then with probability at least $1 - \delta$, we have*

$$\|\mathbf{G}(0) - \mathbf{G}^\infty\|_2 \le \frac{\lambda_0}{4} \quad \text{and} \quad \|\widetilde{\mathbf{G}}(0) - \widetilde{\mathbf{G}}^\infty\|_2 \le \frac{\tilde{\lambda}_0}{4}.$$

To prove this lemma, we need to make some preliminary preparations.

Let $g$ be a non-decreasing function with $g(0) = 0$. The $g$-Orlicz norm of a real-valued random variable $X$ is defined as

$$\|X\|_g := \inf \left\{ t > 0 : \mathbb{E}\left[g\left(\frac{|X|}{t}\right)\right] \le 1 \right\}.$$

A random variable $X$ is said to be sub-Weibull of order $\alpha > 0$, denoted as sub-Weibull($\alpha$), if $\|X\|_{\psi_\alpha} < \infty$, where

$$\psi_\alpha(x) := e^{x^\alpha} - 1, \quad \text{for } x \ge 0.$$

The following result is a commonly used inequality in mathematical fields.

If $X_1, \cdots, X_n$ are independent mean zero random variables with $\|X_i\|_{\psi_\alpha} < \infty$ for all $1 \le i \le n$ and some $\alpha > 0$, then for any vector $a = (a_1, \cdots, a_n) \in \mathbb{R}^n$, the following holds true:

$$P\left(\left|\sum_{i=1}^n a_i X_i\right| \ge 2eC(\alpha)\|b\|_2\sqrt{t} + 2eL_n^*(\alpha)t^{1/\alpha}\|b\|_{\beta(\alpha)}\right) \le 2e^{-t}, \quad \text{for all } t \ge 0, \qquad (38)$$

where $b = (a_1\|X_1\|_{\psi_\alpha}, \cdots, a_n\|X_n\|_{\psi_\alpha}) \in \mathbb{R}^n$,

$$C(\alpha) := \max\left\{\sqrt{2}, \, 2^{1/\alpha}\right\} \begin{cases} \sqrt{8}(2\pi)^{1/4}e^{1/24}(e^{2/e}/\alpha)^{1/\alpha}, & \text{if } \alpha < 1, \\ 4e + 2(\log 2)^{1/\alpha}, & \text{if } \alpha \ge 1, \end{cases}$$

and for $\beta(\alpha) = \infty$ when $\alpha \le 1$ and $\beta(\alpha) = \alpha/(\alpha - 1)$ when $\alpha > 1$,

$$L_n(\alpha) := \frac{4^{1/\alpha}}{\sqrt{2}\|b\|_2} \times \begin{cases} \|b\|_{\beta(\alpha)}, & \text{if } \alpha < 1, \\ 4e\|b\|_{\beta(\alpha)}/C(\alpha), & \text{if } \alpha \ge 1. \end{cases}$$

and $L_n^*(\alpha) = L_n(\alpha)C(\alpha)\|b\|_2/\|b\|_{\beta(\alpha)}$.

*Proof.* We focus on the proof about $\mathbf{G}(0)$.

Since $\|\mathbf{G}(0) - \mathbf{G}^\infty\|_2 \le \|\mathbf{G}(0) - \mathbf{G}^\infty\|_F$, it suffices to bound each entry of $\mathbf{G}(0) - \mathbf{G}^\infty$, which is of the form

$$\sum_{r=1}^m \left\langle \frac{\partial s_p}{\partial w_r}, \frac{\partial s_j}{\partial w_r} \right\rangle - \mathbb{E}_{w \sim \mathcal{N}(0, \mathbf{I}), a \sim \text{Unif}\{-1,1\}} \sum_{r=1}^m \left\langle \frac{\partial s_p}{\partial w_r}, \frac{\partial s_j}{\partial w_r} \right\rangle, \qquad (39)$$

or

$$\sum_{r=1}^m \left\langle \frac{\partial s_p}{\partial w_r}, \frac{\partial h_j}{\partial w_r} \right\rangle - \mathbb{E}_{w \sim \mathcal{N}(0, \mathbf{I}), a \sim \text{Unif}\{-1,1\}} \sum_{r=1}^m \left\langle \frac{\partial s_p}{\partial w_r}, \frac{\partial h_j}{\partial w_r} \right\rangle, \qquad (40)$$

or

$$\sum_{r=1}^m \left\langle \frac{\partial h_p}{\partial w_r}, \frac{\partial h_j}{\partial w_r} \right\rangle - \mathbb{E}_{w \sim \mathcal{N}(0, \mathbf{I}), a \sim \text{Unif}\{-1,1\}} \sum_{r=1}^m \left\langle \frac{\partial h_p}{\partial w_r}, \frac{\partial h_j}{\partial w_r} \right\rangle. \qquad (41)$$

Note that

$$\frac{\partial s_p}{\partial w_r} = \frac{a_r}{\sqrt{mn_1}} \sum_k \sum_{|\xi|=k} c_\xi(y_p^{(1)}) \left[\tanh^{(1+|\xi|)}(w_r^\mathsf{T} y_p^{(1)})w_r^\xi y_p^{(1)} + \tanh^{(|\xi|)}(w_r^\mathsf{T} y_p^{(1)})\frac{\partial w_r^\xi}{\partial w_r}\right]$$

and

$$\frac{\partial h_j}{\partial w_r} = \frac{a_r\sqrt{\lambda}}{\sqrt{mn_2}} \left[\alpha \tanh'(w_r^\mathsf{T} y_j^{(2)})y_j^{(2)} + \beta \tanh''(w_r^\mathsf{T} y_j^{(2)})w_r^\mathsf{T} n(y_j^{(2)}) + \beta \tanh'(w_r^\mathsf{T} y_j^{(2)})n(y_j^{(2)})\right].$$

For the first form Eq.(39), let

$$Y_r(p) = \sum_k \sum_{|\xi|=k} c_\xi(y_p^{(1)}) \left[\tanh^{(1+|\xi|)}(w_r^\mathsf{T} y_p^{(1)})w_r^\xi y_p^{(1)} + \tanh^{(|\xi|)}(w_r^\mathsf{T} y_p^{(1)})\frac{\partial w_r^\xi}{\partial w_r}\right], \quad \forall 1 \le p \le n_1$$

and

$$X_r(ij) = \langle Y_r(i), Y_r(j) \rangle, \quad 1 \le i, j \le n_1.$$

Then we have

$$\sum_{r=1}^{m} \left\langle \frac{\partial s_p}{\partial w_r}, \frac{\partial s_j}{\partial w_r} \right\rangle - \mathbb{E}_{w \sim \mathcal{N}(0,\mathbf{I}), a \sim \mathrm{Unif}\{-1,1\}} \sum_{r=1}^{m} \left\langle \frac{\partial s_p}{\partial w_r}, \frac{\partial s_j}{\partial w_r} \right\rangle = \frac{1}{n_1 m} \sum_{r=1}^{m} (X_r(ij) - \mathbb{E}X_r(ij)).$$

Note that $|X_r(ij)| \lesssim 1 + \|w_r(0)\|_2^{2|\tilde{\xi}|}$, thus

$$\|X_r(ij)\|_{\psi_{\frac{1}{|\tilde{\xi}|}}} \lesssim 1 + \left\| \|w_r(0)\|_2^{2|\tilde{\xi}|} \right\|_{\psi_{\frac{1}{|\tilde{\xi}|}}} \lesssim 1 + \left\| \|w_r(0)\|_2^2 \right\|_{\psi_1}^{|\tilde{\xi}|} \lesssim d^{|\tilde{\xi}|}.$$

For the centered random variable, the property of $\psi_{\frac{1}{|\tilde{\xi}|}}$ quasi-norm implies that

$$\|X_r(ij) - \mathbb{E}[X_r(ij)]\|_{\psi_{\frac{1}{|\tilde{\xi}|}}} \lesssim \|X_r(ij)\|_{\psi_{\frac{1}{|\tilde{\xi}|}}} + \|\mathbb{E}[X_r(ij)]\|_{\psi_{\frac{1}{|\tilde{\xi}|}}} \lesssim d^{|\tilde{\xi}|}.$$

Therefore, applying Eq.(38) (taking $\alpha = \frac{1}{|\tilde{\xi}|}$) yields that with probability at least $1 - \delta$,

$$\left| \frac{1}{m} \sum_{r=1}^{m} (X_r(ij) - \mathbb{E}[X_r(ij)]) \right| \lesssim \frac{d^{|\tilde{\xi}|}}{\sqrt{m}} \sqrt{\log \frac{2}{\delta}} + \frac{d^{|\tilde{\xi}|}}{m} \left( \log \frac{2}{\delta} \right)^{|\tilde{\xi}|},$$

which directly leads to

$$\left| \sum_{r=1}^{m} \left\langle \frac{\partial s_i}{\partial w_r}, \frac{\partial s_j}{\partial w_r} \right\rangle - \mathbb{E}_{(w,a)} \sum_{r=1}^{m} \left\langle \frac{\partial s_i}{\partial w_r}, \frac{\partial s_j}{\partial w_r} \right\rangle \right| \lesssim \frac{d^{|\tilde{\xi}|}}{n_1 \sqrt{m}} \sqrt{\log \frac{2}{\delta}} + \frac{d^{|\tilde{\xi}|}}{n_1 m} \left( \log \frac{2}{\delta} \right)^{|\tilde{\xi}|}.$$

For the second form Eq.(40) and third form Eq.(41), in a similar manner, we can obtain the same result.

Combining the results for the three forms, we can deduce that with probability at least $1 - \delta$,

$$\|\mathbf{G}(0) - \mathbf{G}^\infty\|_2^2 \le \|\mathbf{G}(0) - \mathbf{G}^\infty\|_F^2$$

$$\lesssim \frac{d^{2|\tilde{\xi}|}}{m} \log \frac{2(n_1 + n_2)}{\delta} + \frac{d^{2|\tilde{\xi}|}}{m^2} \left( \log \frac{2(n_1 + n_2)}{\delta} \right)^{2|\tilde{\xi}|}$$

$$\lesssim \frac{d^{2|\tilde{\xi}|}}{m} \log \frac{2(n_1 + n_2)}{\delta}.$$

Thus when $\sqrt{\frac{d^{2|\tilde{\xi}|}}{m} \log \frac{2(n_1+n_2)}{\delta}} \lesssim \frac{\lambda_0}{4}$, i.e.,

$$m = \Omega\left( \frac{d^{2|\tilde{\xi}|}}{\lambda_0^2} \log \left( \frac{n_1 + n_2}{\delta} \right) \right),$$

we have $\lambda_{min}(\mathbf{G}(0)) \ge \frac{3}{4}\lambda_0$.

$\square$

**Lemma 5.** *Let* $R \in (0, 1]$, *if* $w_1(0), \cdots, w_m(0)$ *are i.i.d. generated from* $\mathcal{N}(0, \mathbf{I_{d+1}})$, *then with probability at least* $1 - \delta$, *the following holds. For any set of weight vectors* $\mathbf{W} = (w_1^\mathsf{T}, \ldots, w_m^\mathsf{T})^\mathsf{T}$ *and* $\mathbf{a} = (a_1, \cdots, a_m)^\mathsf{T}$ *satisfying that for any* $1 \le r \le m$, $\|w_r - w_r(0)\|_2 \le R$ *and* $\|\mathbf{a} - \mathbf{a}(0)\|_2 \le R$, *then the induced Gram matrices* $\mathbf{G}(\mathbf{W}, \mathbf{a}), \widetilde{\mathbf{G}}(\mathbf{W}, \mathbf{a})$ *satisfy*

$$\|\mathbf{G}(\mathbf{W}, \mathbf{a}) - \mathbf{G}(0)\|_2 \le CR \left( 1 + \frac{d^{|\tilde{\xi}|}}{\sqrt{m}} \sqrt{\log \frac{2}{\delta}} + \frac{d^{|\tilde{\xi}|}}{m} \left( \log \frac{2}{\delta} \right)^{|\tilde{\xi}|} \right),$$

*and*

$$\|\widetilde{\mathbf{G}}(\mathbf{W}, \mathbf{a}) - \widetilde{\mathbf{G}}(0)\|_2 \le CR \left( 1 + \frac{d^{|\tilde{\xi}|}}{\sqrt{m}} \sqrt{\log \frac{2}{\delta}} + \frac{d^{|\tilde{\xi}|}}{m} \left( \log \frac{2}{\delta} \right)^{|\tilde{\xi}|} \right)$$

*where* $C$ *is a universal constant.*

*Proof.* As $\|\mathbf{G}(\mathbf{W}, \mathbf{a}) - \mathbf{G}(0)\|_2 \leq \|\mathbf{G}(\mathbf{W}, \mathbf{a}) - \mathbf{G}(0)\|_F$, it suffices to bound each entry.

Note that

$$\frac{\partial s_p}{\partial w_r} = \frac{a_r}{\sqrt{mn_1}} \sum_k \sum_{|\xi|=k} c_\xi(y_p^{(1)}) \left[ \tanh^{(1+|\xi|)}(w_r^\top y_p^{(1)}) w_r^\xi y_p^{(1)} + \tanh^{(|\xi|)}(w_r^\top y_p^{(1)}) \frac{\partial w_r^\xi}{\partial w_r} \right]$$

and

$$\frac{\partial h_j}{\partial w_r} = \frac{a_r \sqrt{\lambda}}{\sqrt{mn_2}} \left[ \alpha \tanh'(w_r^\top y_j^{(2)}) y_j^{(2)} + \beta \tanh''(w_r^\top y_j^{(2)}) w_r^\top n(y_j^{(2)}) + \beta \tanh'(w_r^\top y_j^{(2)}) n(y_j^{(2)}) \right].$$

For $1 \leq i, j \leq n_1$, noticing that all higher-order derivatives of $\tanh$ are bounded and $R \in (0, 1]$, we have

$$|G_{ij}(\mathbf{W}, \mathbf{a}) - G_{ij}(0)|$$

$$= \left| \sum_{r=1}^m \left\langle \frac{\partial s_i(\mathbf{W}, \mathbf{a})}{\partial w_r}, \frac{\partial s_j(\mathbf{W}, \mathbf{a})}{\partial w_r} \right\rangle - \left\langle \frac{\partial s_i(\mathbf{W}(0), \mathbf{a}(0))}{\partial w_r}, \frac{\partial s_j(\mathbf{W}(0), \mathbf{a}(0))}{\partial w_r} \right\rangle \right|$$

$$\lesssim R \frac{1}{n_1 m} \sum_{r=1}^m (\|w_r(0)\|_2^{2|\tilde{\xi}|} + 1).$$

For $1 \leq i \leq n_1, n_1 + 1 \leq j \leq n_1 + n_2$, we also have

$$|G_{ij}(\mathbf{W}, \mathbf{a}) - G_{ij}(0)|$$

$$= \left| \sum_{r=1}^m \left\langle \frac{\partial s_i(\mathbf{W}, \mathbf{a})}{\partial w_r}, \frac{\partial h_j(\mathbf{W}, \mathbf{a})}{\partial w_r} \right\rangle - \left\langle \frac{\partial s_i(\mathbf{W}(0), \mathbf{a}(0))}{\partial w_r}, \frac{\partial h_j(\mathbf{W}(0), \mathbf{a}(0))}{\partial w_r} \right\rangle \right|$$

$$\lesssim R \frac{1}{\sqrt{n_1 n_2} m} \sum_{r=1}^m (\|w_r(0)\|_2^{2|\tilde{\xi}|} + 1).$$

For $n_1 + 1 \leq i, j \leq n_1 + n_2$, we still have

$$|G_{ij}(\mathbf{W}, \mathbf{a}) - G_{ij}(0)|$$

$$= \left| \sum_{r=1}^m \left\langle \frac{\partial h_i(\mathbf{W}, \mathbf{a})}{\partial w_r}, \frac{\partial h_j(\mathbf{W}, \mathbf{a})}{\partial w_r} \right\rangle - \left\langle \frac{\partial h_i(\mathbf{W}(0), \mathbf{a}(0))}{\partial w_r}, \frac{\partial h_j(\mathbf{W}(0), \mathbf{a}(0))}{\partial w_r} \right\rangle \right|$$

$$\lesssim R \frac{1}{n_2 m} \sum_{r=1}^m (\|w_r(0)\|_2^2 + 1)$$

$$\lesssim R \frac{1}{n_2 m} \sum_{r=1}^m (\|w_r(0)\|_2^{2|\tilde{\xi}|} + 1).$$

Combining above results yields that

$$\|\mathbf{G}(\mathbf{W}, \mathbf{a}) - \mathbf{G}(0)\|_2^2 \leq \|\mathbf{G}(\mathbf{W}, \mathbf{a}) - \mathbf{G}(0)\|_F^2 \lesssim R^2 + R^2 \left( \frac{1}{m} \sum_{r=1}^m \|w_r(0)\|_2^{|\tilde{\xi}|} \right)^2.$$

For the second term, applying Eq.(38) implies that with probability at least $1 - \delta$,

$$\frac{1}{m} \sum_{r=1}^m \|w_r(0)\|_2^{2|\tilde{\xi}|} \lesssim \frac{d^{|\tilde{\xi}|}}{\sqrt{m}} \sqrt{\log \frac{2}{\delta}} + \frac{d^{|\tilde{\xi}|}}{m} \left( \log \frac{2}{\delta} \right)^{|\tilde{\xi}|}. \tag{42}$$

Finally, we can deduce that with probability at least $1 - \delta$,

$$\|\mathbf{G}(\mathbf{W}, \mathbf{a}) - \mathbf{G}(0)\|_2 \leq CR \left( 1 + \frac{d^{|\tilde{\xi}|}}{\sqrt{m}} \sqrt{\log \frac{2}{\delta}} + \frac{d^{|\tilde{\xi}|}}{m} \left( \log \frac{2}{\delta} \right)^{|\tilde{\xi}|} \right),$$

where $C$ is a universal constant. $\qquad \square$

**Lemma 6** (Bounded initial loss). *With probability at least $1 - \delta$, we have*

$$\mathcal{J}_{emp}(0) \leq C \left( d^{2|\tilde{\xi}|} \log \left( \frac{n_1 + n_2}{\delta} \right) + \frac{d^{2|\tilde{\xi}|}}{m} \left( \log \left( \frac{n_1 + n_2}{\delta} \right) \right)^{2|\tilde{\xi}|} \right), \quad (43)$$

*where $C$ is a universal constant.*

*Proof.* For the initial value of PINN, we have

$$\mathcal{J}_{\text{emp}}(0) = \frac{1}{2} \sum_{p=1}^{n_1} s_p^2(\mathbf{W}(0), \mathbf{a}(0)) + \frac{1}{2} \sum_{j=1}^{n_2} h_j^2(\mathbf{W}(0), \mathbf{a}(0))$$

$$= \frac{1}{2n_1} \sum_{p=1}^{n_1} \left( \frac{1}{\sqrt{m}} \sum_{r=1}^{m} a_r(0) \sum_{k=0}^{+\infty} \sum_{|\xi|=k} c_\xi(y_p^{(1)}) \tanh^{(|\xi|)}(w_r(0)^\mathsf{T} y_p^{(1)}) w^\xi - f(y_p^{(1)}) \right)^2$$

$$+ \frac{1}{2n_2} \sum_{j=1}^{n_2} \left( \frac{1}{\sqrt{m}} \sum_{r=1}^{m} a_r(0)(\alpha \tanh(w^T y_j^{(2)}) + \beta \tanh'(w^\mathsf{T} y_j^{(2)}) w^\mathsf{T} n(y_j^{(2)})) - g(y_j^{(2)}) \right)^2$$

$$\leq \frac{1}{n_1} \sum_{p=1}^{n_1} \left( \frac{1}{\sqrt{m}} \sum_{r=1}^{m} a_r(0) \sum_{k=0}^{+\infty} \sum_{|\xi|=k} c_\xi(y_p^{(1)}) \tanh^{(|\xi|)}(w_r(0)^\mathsf{T} y_p^{(1)}) w^\xi \right)^2 + (f(y_p^{(1)}))^2$$

$$+ \frac{1}{n_2} \sum_{j=1}^{n_2} \left( \frac{1}{\sqrt{m}} \sum_{r=1}^{m} a_r(0)\alpha \tanh(w^T y_j^{(2)}) \right)^2$$

$$+ \frac{1}{n_2} \sum_{j=1}^{n_2} \left( \frac{1}{\sqrt{m}} \sum_{r=1}^{m} a_r(0)\beta \tanh'(w^\mathsf{T} y_j^{(2)}) w^\mathsf{T} n(y_j^{(2)}) \right)^2 + (g(y_j^{(2)}))^2.$$

$$(44)$$

For the first term in Eq.(44), note that $\mathbb{E}\left[ a_r(0) \sum_{k=0}^{+\infty} \sum_{|\xi|=k} c_\xi(y_p^{(1)}) \tanh^{(|\xi|)}(w_r(0)^\mathsf{T} y_p^{(1)}) w^\xi \right] = 0$ and

$$\left| a_r(0) \sum_{k=0}^{+\infty} \sum_{|\xi|=k} c_\xi(y_p^{(1)}) \tanh^{(|\xi|)}(w_r(0)^\mathsf{T} y_p^{(1)}) w^\xi \right| \lesssim 1 + \|w_r(0)\|_2^{|\tilde{\xi}|}.$$

Therefore, we have

$$\left\| a_r(0) \sum_{k=0}^{+\infty} \sum_{|\xi|=k} c_\xi(y_p^{(1)}) \tanh^{(|\xi|)}(w_r(0)^\mathsf{T} y_p^{(1)}) w^\xi \right\|_{\psi_{\frac{1}{|\tilde{\xi}|}}} \lesssim 1 + \left\| \|w_r(0)\|_2^{|\tilde{\xi}|} \right\|_{\psi_{\frac{1}{|\tilde{\xi}|}}} \lesssim d^{|\tilde{\xi}|}.$$

Let $X_r = a_r(0) \sum_{k=0}^{+\infty} \sum_{|\xi|=k} c_\xi(y_p^{(1)}) \tanh^{(|\xi|)}(w_r(0)^\mathsf{T} y_p^{(1)}) w^\xi$, then with probability at least $1 - \delta$,

$$\left| \sum_{r=1}^{m} \frac{X_r}{\sqrt{m}} \right| \lesssim d^{|\tilde{\xi}|} \sqrt{\log \frac{2}{\delta}} + \frac{d^{|\tilde{\xi}|}}{\sqrt{m}} \left( \log \frac{2}{\delta} \right)^{|\tilde{\xi}|}.$$

As for the second term, we have $\mathbb{E}[a_r(0)\alpha \tanh(w_r(0)^T y_j^{(2)})] = 0$ and by Lipschitz continuty,

$$|a_r(0)\alpha \tanh(w_r(0)^T y_j^{(2)})| \lesssim |w_r(0)^T y_j^{(2)}|.$$

Thus

$$\|a_r(0)\alpha \tanh(w_r(0)^T y_j^{(2)})\|_{\psi_2} \leq C,$$

as $w_r(0)^T y_j^{(2)} \sim \mathcal{N}(0, \|y_j^{(2)}\|_2^2)$.

Let $Y_r = a_r(0)\alpha \tanh(w_r(0)^T y_j^{(2)})$, applying Eq.(38) yields that with probability at least $1 - \delta$,

$$\left| \sum_{r=1}^m \frac{Y_r}{\sqrt{m}} \right| \lesssim \sqrt{\log\left(\frac{1}{\delta}\right)} + \frac{1}{\sqrt{m}} \log\left(\frac{1}{\delta}\right).$$

Finally, similar to the approach used for the first term, we can also control the third term. Let $Z_r = a_r(0)\beta \tanh'(w^\intercal y_j^{(2)}) w^\intercal n(y_j^{(2)})$, then with probability at least $1 - \delta$,

$$\left| \sum_{r=1}^m \frac{Z_r}{\sqrt{m}} \right| \lesssim d\sqrt{\log\frac{2}{\delta}} + \frac{d}{\sqrt{m}} \log\frac{2}{\delta}.$$

Combining all results above yields that

$$\mathcal{J}_{\text{emp}}(0) \lesssim d^{2|\tilde{\xi}|} \log\left(\frac{n_1 + n_2}{\delta}\right) + \frac{d^{2|\tilde{\xi}|}}{m} \left(\log\left(\frac{n_1 + n_2}{\delta}\right)\right)^{2|\tilde{\xi}|}$$

holds with probability at least $1 - \delta$. $\qquad\square$

**Lemma 7.** *With probability at least* $1 - \delta$,

$$\|w_r(0)\|_2^2 \le C\left(d + \sqrt{d\log\left(\frac{m}{\delta}\right)} + \log\left(\frac{m}{\delta}\right)\right) := R'^2 \qquad (45)$$

*holds for all* $1 \le r \le m$ *and* $C$ *is a universal constant.*

*Proof.* From Eq.(38), we can deduce that for fixed $r$,

$$\|w_r(0)\|_2^2 \le C\left(d + \sqrt{d\log\left(\frac{1}{\delta}\right)} + \log\left(\frac{1}{\delta}\right)\right)$$

holds with probability at least $1 - \delta$.

Therefore, the following holds with probability at least $1 - \delta$.

$$\|w_r(0)\|_2^2 \le C\left(d + \sqrt{d\log\left(\frac{m}{\delta}\right)} + \log\left(\frac{m}{\delta}\right)\right), \; \forall r \in [m].$$

$\square$

**Lemma 8.** *Let* $R = \mathcal{O}\left(\frac{\min\{\lambda_0, \tilde{\lambda}_0\}}{d^{|\tilde{\xi}|}(\log\frac{2}{\delta})^{|\tilde{\xi}|}}\right)$. *If*

$$m = \Omega\left(\frac{1}{\left(\lambda_0 + \tilde{\lambda}_0\right)^2} \cdot d^{2|\tilde{\xi}|} \left(\log\left(\frac{n_1 + n_2}{\delta}\right)\right)^{2|\tilde{\xi}|} \cdot \frac{R'^6}{R^2}\right),$$

*and assuming* $|a_r(\tau)| \le 2$, $\|w_r(\tau)\|_2 \le 2R'$, $\lambda_{\min}(G(\mathbf{W}(\tau), \mathbf{a}(\tau))) \ge \frac{\lambda_0}{2}$, *and* $\lambda_{\min}(\widetilde{G}(\mathbf{W}(\tau), \mathbf{a}(\tau))) \ge \frac{\tilde{\lambda}_0}{2}$ *for all* $0 \le \tau \le t$, *then* $\|w_r(\tau) - w_r(0)\|_2 \le R$ *and* $|a_r(\tau) - a_r(0)| \le R$ *for all* $r \in [m]$ *and* $0 \le \tau \le t$.

*Proof.* The proof follows the same approach as Lemma B.2 in the paper Gao et al. (2023) and is omitted here for brevity. $\qquad\square$

After the preparation of the previous lemmas, we now present the complete proof of Theorem 1.

*Proof.* Finally, for all $r \in [m]$, $w_r(t)$ and $\mathbf{a}(t)$ will remain within the balls $B(w_r(0), R)$ and $B(\mathbf{a}(0), R)$, respectively. Without loss of generality, let us assume $R' \geq R$, so that $\|w_r(\tau)\|_2 \leq 2R'$ if $w_r(\tau)$ stays inside $B(w_r(0), R)$. Lemma 8 shows that if

$$m = \Omega \left( \frac{1}{\left( \lambda_0 + \tilde{\lambda}_0 \right)^2} \cdot d^{2|\tilde{\xi}|} \left( \log \left( \frac{n_1 + n_2}{\delta} \right) \right)^{2|\tilde{\xi}|} \cdot \frac{R'^6}{R^2} \right)$$

$$= \widetilde{\Omega} \left( \frac{1}{\left( \lambda_0 + \tilde{\lambda}_0 \right)^2} \cdot d^{4|\tilde{\xi}|} \left( \log \left( \frac{n_1 + n_2}{\delta} \right) \right)^{4|\tilde{\xi}|} \cdot \frac{d^3}{\min\{\lambda_0^2, \tilde{\lambda}_0^2\}} \right),$$

then for all $t > 0$ and $1 \leq r \leq m$, we have $\|w_r(t) - w_r(0)\|_2 \leq R$, $\|\mathbf{a}(t) - \mathbf{a}(0)\|_2 \leq R$, $\lambda_{\min}(G(\mathbf{W}(t), \mathbf{a}(t))) \geq \frac{\lambda_0}{2}$, and $\lambda_{\min}(\widetilde{G}(\mathbf{W}(t), \mathbf{a}(t))) \geq \frac{\tilde{\lambda}_0}{2}$. Then we have

$$\frac{\mathrm{d}\mathcal{J}_{\mathrm{emp}}(\mathbf{W}(t), \mathbf{a}(t))}{\mathrm{d}t} = \frac{1}{2} \frac{\mathrm{d}}{\mathrm{d}t} \left\| \begin{pmatrix} s(\mathbf{W}(t), \mathbf{a}(t)) \\ h(\mathbf{W}(t), \mathbf{a}(t)) \end{pmatrix} \right\|_2^2$$

$$= - \left[ s(\mathbf{W}(t), \mathbf{a}(t))^\top, h(\mathbf{W}(t), \mathbf{a}(t))^\top \right] \cdot \left( \mathbf{G}(\mathbf{W}(t), \mathbf{a}(t)) + \widetilde{\mathbf{G}}(\mathbf{W}(t), \mathbf{a}(t)) \right) \cdot \begin{pmatrix} s(\mathbf{W}(t), \mathbf{a}(t)) \\ h(\mathbf{W}(t), \mathbf{a}(t)) \end{pmatrix}$$

$$\leq -\frac{1}{2}(\lambda_0 + \tilde{\lambda}_0) \cdot \left\| \begin{pmatrix} s(\mathbf{W}(t), \mathbf{a}(t)) \\ h(\mathbf{W}(t), \mathbf{a}(t)) \end{pmatrix} \right\|_2^2$$

$$= -(\lambda_0 + \tilde{\lambda}_0) \cdot \mathcal{J}_{\mathrm{emp}}(\mathbf{W}(t), \mathbf{a}(t)).$$

Furthermore,

$$\mathcal{J}_{\mathrm{emp}}(\mathbf{W}(t), \mathbf{a}(t)) \leq \exp \left( -(\lambda_0 + \bar{\lambda}_0) \cdot t \right) \cdot \mathcal{J}_{\mathrm{emp}}(w(0), a(0)),$$

for all $t > 0$. $\square$

## H EXTENSION TO DEEPER NETWORKS

In this section, we discuss how our main results (Theorem 1 and Theorem 3) can be extended to deeper neural network architectures. For linear PDEs, the convergence analysis grounded in neural tangent kernel (NTK) theory naturally generalizes to multi-layer networks, leveraging established results from the NTK literature. In the context of nonlinear PDEs, we address the critical issue of the linear independence of neuron functions (Lemma 1) and summarize recent theoretical advances that provide sufficient conditions for preserving this property in deeper networks, especially three-layer architectures. The relevant literature and further details are reviewed below.

### H.1 CONVERGENCE RESULTS FOR SOLVING LINEAR PDES WITH DEEPER NEURAL NETWORKS

Previous works (Gao et al., 2023; Li et al., 2023b) on the convergence of PINN frameworks for second-order elliptic equations, both for gradient descent and implicit gradient descent, have been primarily limited to two-layer neural networks. In contrast, Du et al. (2019b) establishes convergence of the loss function for over-parameterized, multi-layer fully connected networks in the supervised learning setting, fundamentally relying on the neural tangent kernel (NTK) theory for deep networks. By combining the proof strategy of our result Theorem 1 with the layer-wise NTK analysis from Du et al. (2019b), the convergence guarantees within the PINN framework can be extended to over-parameterized, multi-layer fully connected networks, provided either implicit gradient descent or gradient descent with a sufficiently small step size is used.

Below, we state an informal theorem (omitting explicit over-parameterization bounds), as in practice it suffices to select the network width large enough to observe the convergence behavior, rather than strictly adhering to theoretical minima.

**Theorem 6** (Informal: Convergence of Multi-layer PINNs for Certain Linear PDEs). *Consider a physics-informed neural network (PINN) with a deep (multi-layer) architecture used to approximate*

*the solution of an admissible linear PDE, where the empirical loss is defined analogously to Eq.(6), and assume standard random initialization for all weights and biases. Under gradient flow training and with a sufficiently wide network, the empirical loss $\mathcal{J}_{\mathrm{emp}}(\theta(t))$ decreases to zero as $t \to \infty$, with a convergence rate governed by the spectrum of the neural tangent kernel associated with the multi-layer architecture.*

Furthermore, Du et al. (2019b) also extends convergence analyses to convolutional and ResNet architectures in the supervised learning setting. This indicates that similar convergence results for PINNs could potentially be obtained for these more advanced architectures, which is a promising direction for future research.

## H.2 CONVERGENCE RESULTS FOR SOLVING NONLINEAR PDEs WITH THREE-LAYER NEURAL NETWORKS

To the best of our knowledge, our work is the first to investigate the convergence of PINNs for solving nonlinear PDEs, even though it is restricted to the two-layer random feature model. Following the structure of Section 4.2 in the main text, we note that the central step in the proof of Theorem 3 is to establish the coercivity of the loss function with respect to the trainable parameter $a$. This step fundamentally depends on demonstrating the linear independence of the neuron basis functions (see Proposition 2), which, in turn, relies on Lemma 1.

Extending Theorem 3 to multi-layer random feature models—where all hidden layer parameters are fixed randomly—thus essentially reduces to ensuring that Lemma 1 holds for multi-layer architectures. In this context, the recent work Zhang (2024) provides a relevant discussion and establishes the following result.

**Proposition 5** (Proposition 5.3 in Zhang (2024)). *Given $d, m, n \in \mathbb{N}$. Let $\{(w_k^{(1)}, b_k^{(1)})\}_{k=1}^m \subset \mathbb{R}^{md+m}$ be such that $(w_{k_1}^{(1)}, b_{k_1}^{(1)}) \pm (w_{k_2}^{(1)}, b_{k_2}^{(1)}) \neq 0$ for all distinct $k_1, k_2 \in \{1, \ldots, m\}$ and $w_k^{(1)} \neq 0$ for all $k \in \{1, \ldots, m\}$. Let $\{(w_j^{(2)}, b_j^{(2)})\}_{j=1}^n \subset \mathbb{R}^{mn+n}$ be such that $(w_{j_1}^{(2)}, b_{j_1}^{(2)}) \pm (w_{j_2}^{(2)}, b_{j_2}^{(2)}) \neq 0$ for all distinct $j_1, j_2 \in \{1, \ldots, n\}$ and $w_j^{(2)} \neq 0$ for all $j \in \{1, \ldots, n\}$. Then for $\sigma$ being a sigmoid or tanh activation function, the three-layer neurons*

$$\left\{ \sigma \left( \sum_{k=1}^m w_{jk}^{(2)} \, \sigma \left( w_k^{(1)} z + b_k^{(1)} \right) + b_j^{(2)} \right) \right\}_{j=1}^n$$

*are linearly independent.*

Although the three-layer result is more general than our Lemma 1, the proof of Lemma 1 is much more straightforward, while the three-layer result relies on elaborate arguments in the cited work. Therefore, by applying the above result and following the proof strategy of Theorem 3, we can obtain the following convergence theorem.

**Random Initialization (Three-layer Network)** Inner-layer parameters $\{(w_k^{(1)}, b_k^{(1)})\}_{k=1}^m$ and $\{(w_j^{(2)}, b_j^{(2)})\}_{j=1}^n$ are randomly initialized as follows:

$$w_k^{(1)} \sim \mathcal{N}(0, \mathrm{Id}) \text{ i.i.d.}, \quad w_j^{(2)} \sim \mathcal{N}(0, \mathrm{Id}) \text{ i.i.d.},$$

$$b_k^{(1)} \sim \mathcal{N}(0, 1) \text{ i.i.d.}, \quad b_j^{(2)} \sim \mathcal{N}(0, 1) \text{ i.i.d.}$$

for $1 \leq k \leq m$, $1 \leq j \leq n$.

**Theorem 7** (Almost sure convergence via admissible initialization for three-layer networks). *Under Assumption 1, regardless of the specific form of the differential operator $\mathcal{L}$ in the PDE with Dirichlet boundary condition, we can initialize the inner parameters $\{(w_k^{(1)}, b_k^{(1)})\}_{k=1}^m$ and $\{(w_j^{(2)}, b_j^{(2)})\}_{j=1}^n$ with probability 1 such that:*

*(i) $\mathcal{J}$ is coercive about $a$;*

*(ii) All convergence results of Proposition 1 hold for the three-layer setting.*

As for more complex network architectures, we are currently unable to provide a rigorous theoretical result, and this will be the subject of future research. It is also worth noting that when the inner-layer parameters are trainable, even in the two-layer case, current theory only guarantees either divergence to infinity or convergence to a critical point. Without imposing additional assumptions, it is not yet possible to rule out parameter divergence, and thus convergence cannot be ensured.

# I   ADDITIONAL EXPERIMENTS

In this section, we provide experimental details and results to supplement the main text. We begin by presenting the pseudocode for implementing PINNs with implicit gradient descent, followed by a detailed description of the experimental setup for the Burgers' equation. Finally, we provide further experimental setups and results for high-dimensional test problems. In addition, all experiments in this paper were conducted on a desktop computer equipped with a single 4060Ti GPU.

## I.1   PSEUDOCODE FOR IGD

Compared to standard optimization algorithms such as gradient descent or stochastic gradient descent, implicit gradient descent is less commonly used and may be less familiar to readers. Therefore, we provide a detailed explanation here. Let $\theta$ denote all trainable parameters in the network and $\mathcal{J}(\theta)$ represent the empirical loss function. The iteration rule for implicit gradient descent is given by:

$$\theta^{k+1} = \theta^k - \eta \nabla \mathcal{J}(\theta^{k+1}), \quad k = 0, 1, 2, \ldots.$$

This update step can be interpreted as solving the following optimization problem:

$$\min_{\xi} \ \eta \mathcal{J}(\xi) + \frac{1}{2}\|\xi - \theta^k\|_2^2. \tag{46}$$

The first-order optimality condition for this problem is equivalent to the IGD update rule. Consequently, regardless of the step size, the parameter sequence generated by IGD guarantees a monotonically decreasing loss value. This inherent stability allows IGD to impose much weaker restrictions on the choice of step size compared to standard gradient descent.

It is important to note that the operator $I + \eta \nabla \mathcal{J}$ may not be invertible for arbitrary loss functions. However, when $\mathcal{J}$ is convex, proximal point theory ensures that the subproblem above always admits a solution for any step size $\eta$. In practice, PINNs often employ second-order solvers such as L-BFGS to efficiently solve the subproblem Eq.(46), even when convexity is not strictly satisfied. This practical approach makes IGD a robust and effective choice for real-world applications. The pseudocode for implementing the IGD algorithm is provided below in Algorithm 1.

---

**Algorithm 1** Mini-batch implicit gradient descent (IGD)

---

1: **Input:**
   Training dataset $\mathcal{D}$; Batch size $B$;
   Number of outer iterations $K_1$; Number of inner iterations $K_2$;
   Outer (IGD) step size $\eta$; Inner (solver) step size $\gamma$; Initial parameters $\theta^0$.
2: **for** $k = 0$ **to** $K_1 - 1$ **do**
3:    Sample a batch $\mathcal{B}_k$ of size $B$ from $\mathcal{D}$
4:    Define empirical loss function $\mathcal{J}_k(\theta)$ on $\mathcal{B}_k$
5:    **Inner loop:**
      Use L-BFGS optimizer to approximately solve Eq.(46) with $K_2$ iterations and (internal) step size $\gamma$.
      Let $\xi^0 = \theta^k$.
6:    **for** $t = 0$ **to** $K_2 - 1$ **do**
7:       $\xi^{t+1}$ is obtained by applying one L-BFGS step to $\xi^t$ on the objective in Eq.(46).
8:    **end for**
9:    Set $\theta^{k+1} = \xi^{K_2}$ as the output of the inner loop
10: **end for**
11: **Output:** Final parameters $\theta^{K_1}$

---

## I.2 EXPERIMENT SETUP FOR BURGERS' EQUATION

In the main text, for the sake of brevity, we only presented the results of our partial experiments. Here, to ensure the reliability and reproducibility of our findings, we provide comprehensive details of the experimental setup, including the specific hyperparameter choices.

**Experiment setup for Figure 1.** The primary goal of Figure 1 is to illustrate that the neural tangent kernel (NTK) matrix induced by the loss corresponding to the nonlinear differential operator evolves significantly during training, while the NTK matrix corresponding to the linear boundary operator remains nearly unchanged. To demonstrate this effect, we used a scaled two-layer neural network as the model architecture:

$$u(t, x; \theta) = \frac{1}{\sqrt{1000}} \sum_{k=1}^{1000} a_k \tanh\left(w_k^\top (t, x)\right).$$

The weight parameters $w_k$ were initialized using the standard normal distribution, while the outer parameters $a_k$ were initialized uniformly over the interval $[-1, 1]$. The training dataset consisted of 100 interior points and 20 boundary points. The relatively small number of data points, compared to the network width, was chosen to satisfy the overparameterization conditions assumed in NTK theory. Throughout training, no mini-batching was used; instead, full-batch updates were performed at every iteration.

The implicit gradient descent (IGD) algorithm was used to optimize the network parameters $a = (a_k)_{1 \leq k \leq 1000}$, while the weights $w = (w_k)_{1 \leq k \leq 1000}$ were kept fixed throughout training. The outer IGD iterations were performed for 100 steps with a step size of $\eta = 0.5$. At each outer iteration, the inner subproblem Eq.(46) was solved using the L-BFGS optimizer, with a step size of 0.1 and 10 iterations per outer step. The result shown in Figure 1 was generated under these settings. A summary of the chosen hyperparameters is provided in Table 2 for reference.

Table 2: Hyperparameter settings for the experiment in Figure 1.

| Component | Value | Description |
|---|---|---|
| Interior points | 100 | Training data points (domain) |
| Boundary points | 20 | Training data points (boundary) |
| Batching | Full | No mini-batch |
| Outer IGD steps | 100 | Total optimization iterations |
| IGD step size | 0.5 | Step size for outer loop |
| Inner solver | L-BFGS | Optimizer for each IGD step |
| L-BFGS steps | 10 | Inner iterations per IGD step |
| L-BFGS step size | 0.1 | Step size for L-BFGS |

**Experiment setup for convergence validation on Burgers' equation.** This experiment aims to empirically validate Theorem 3, which concerns the convergence of the random feature model when solving nonlinear equations. The neural network used is a two-layer model with width 100:

$$u(t, x; \theta) = \sum_{k=1}^{100} a_k \tanh\left(w_k^\top (t, x)\right).$$

The weights $w_k$ were initialized from a standard normal distribution, while the coefficients $a_k$ were initialized uniformly in the interval $[-1, 1]$. The training dataset contains 10,000 interior points and 100 boundary points, clearly not in an overparameterized regime. Full-batch training is employed for this experiment.

IGD algorithm is used to optimize the outer-layer parameters. The step size of the outer iteration is $\eta = 0.5$ for 10000 steps. Each IGD subproblem is approximately solved using the L-BFGS optimizer (20 inner steps per outer loop) with a step size of 0.01. A summary of the key settings is provided in Table 3.

Table 3: Hyperparameter settings of the experiment for convergence validation on Burgers' equation.

| Component | Value |
|---|---|
| Parameter initialization | $w_k \sim \mathcal{N}(0, \mathbf{I}_2), a_k \sim \mathcal{U}[-1, 1]$ |
| Interior points | 10,000 |
| Boundary points | 100 |
| Batching | Full batch |
| Optimizer (IGD) | Outer steps: 10000, step size $\eta = 0.5$ |
| Inner solver (IGD) | L-BFGS, 20 steps/outer step, step size $= 0.01$ |

### I.3 HIGH-DIMENSIONAL EXPERIMENTS

To further validate the theoretical results presented in the main text, we conduct experiments on high-dimensional nonlinear partial differential equations. In particular, we consider both the Allen–Cahn and Fisher–KPP equations as representative examples. These experiments are designed to test whether our theoretical insights hold in more challenging, high-dimensional scenarios. The detailed settings and results for each equation are presented in the following subsections.

#### I.3.1 ALLEN–CAHN EQUATION

We consider the two-dimensional Allen–Cahn equation,

$$u_t = \epsilon^2 \Delta u - (u^3 - u) + S(x, y, t), \tag{47}$$

on $(x, y) \in [-1, 1] \times [-1, 1]$, $t \in [0, 1]$, with $\epsilon = 0.1$. The exact solution we set is

$$u(x, y, t) = [\sin(\pi x)\cos(\pi y) + 0.1 \sin(10\pi x)\cos(10\pi y)]e^{-t},$$

from which $S(x, y, t)$, initial, and boundary conditions are determined.

**Experiment 1: NTK failure in the random feature model for nonlinear PDEs.**

To solve Eq.(47) within the PINN framework, we use a shallow neural network of the form

$$u_\theta = \frac{1}{\sqrt{1000}} \sum_{k=1}^{1000} a_k \tanh\left(w_k^\top \mathbf{x} + b_k\right),$$

where $\mathbf{x} = (x, y, t)^\top$. The inner parameters $(w_k, b_k)$ are initialized as standard Gaussian and kept fixed, while the outer coefficients $a_k$ are initialized uniformly in $(-1, 1)$ and optimized by the IGD algorithm. The dataset consists of 50 boundary and 200 interior points, with full-batch training. Optimization is performed for 100 outer steps of step size 0.5 (each with 10 L-BFGS inner iterations of step size 0.1). We track the relative Frobenius norm of the NTK matrices during training, as shown in Figure 2. The results indicates that while the NTK remains stable for the linear (boundary) operator, it changes significantly for the nonlinear (interior) operator, reflecting a breakdown of the NTK regime even in the random feature model for nonlinear problems.

**Experiment 2: Convergence in the random feature model.**

In this experiment, we continue to use the random feature model $u_\theta = \sum_{k=1}^{1000} a_k \tanh(w_k^\top \mathbf{x} + b_k)$, with $(w_k, b_k)$ fixed after Gaussian initialization and $a_k$ initialized uniformly in $(-1, 1)$ and optimized by IGD. The dataset contains 500 boundary and 10,000 interior points. Training is performed in full batch for 2,000,000 total steps (50,000 IGD outer steps with step size 0.5, each with 40 L-BFGS inner steps of step size 0.1).

At the end of training, the $\ell_2$-norm of the loss gradient with respect to parameters is about $1.86 \times 10^{-3}$, indicating convergence to a critical point, which is consistent with our theoretical analysis in Theorem 3. We note that the norm is not zero, likely due to the problem's multiscale nature and the finite training budget: even 2,000,000 steps are sometimes insufficient for full convergence in such stiff problems (prior works have reported using up to 5,000,000 steps).

**Experiment 3: IGD outperforms Adam on multi-scale problems with large step sizes.**

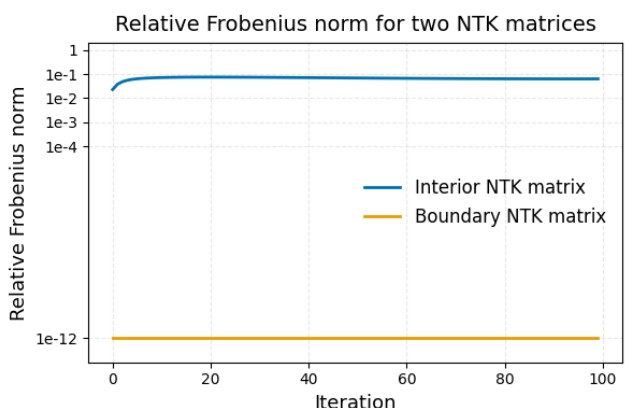

Figure 2: Relative Frobenius norm of two NTK matrices for Allen–Cahn equation.

We further assess the performance of IGD and Adam on a more expressive model: a fully connected neural network comprising four layers with 100 neurons each. Biases are initialized to zero, and all other weights are initialized using Xavier normal initialization. Both IGD and Adam employ the same initialization scheme. The training data contains 500 boundary points and 20,000 interior points, with mini-batch sizes of 32 and 256 for boundary and interior points, respectively.

Both optimizers are trained for a total of 1,000,000 steps. For IGD, this corresponds to 25,000 outer steps (learning rate 0.1), with each outer step followed by 40 L-BFGS inner iterations (learning rate 0.1). Adam uses a constant learning rate of 0.1 throughout all iterations.

Figure 3 shows the loss curves over the entire training process for both IGD and Adam algorithms. With a step size of 0.1, IGD exhibits steady and stable loss reduction, whereas Adam experiences severe oscillations and fails to make substantive progress on this multiscale problem. These results further underscore the robustness and effectiveness of IGD in challenging multiscale settings. Furthermore, Figure 4 shows the solutions obtained by IGD at three representative time points. As shown, the learned solution captures some key features of the ground truth.

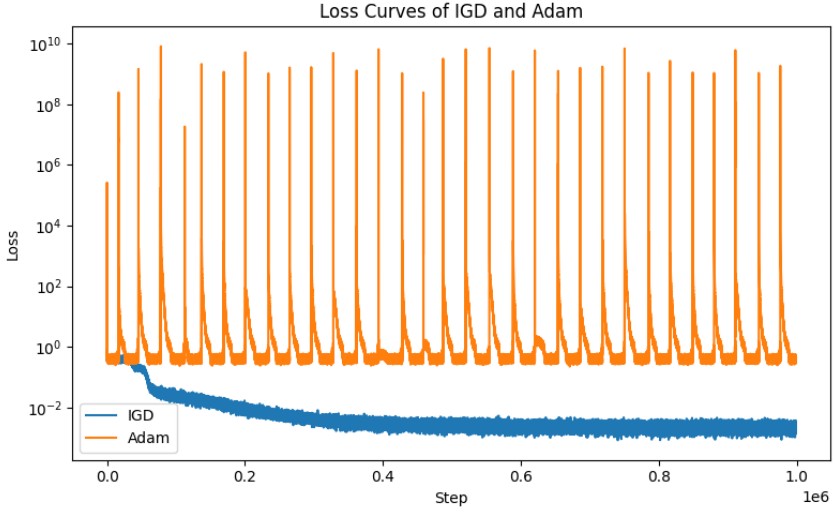

Figure 3: Loss curves for IGD and Adam on Allen–Cahn equation.

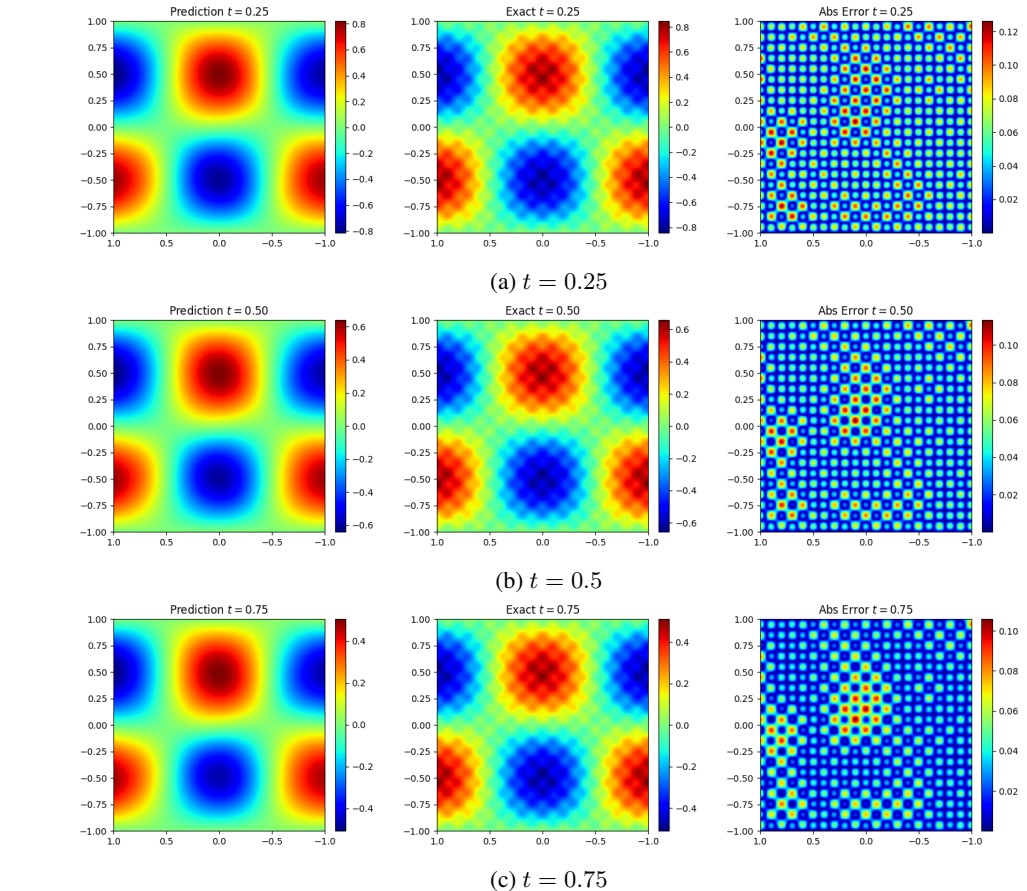

(a) $t = 0.25$

(b) $t = 0.5$

(c) $t = 0.75$

Figure 4: Numerical solutions obtained by IGD at 3 different time points for Allen–Cahn equation.

### I.3.2 FISHER–KPP EQUATION

To further supplement our main results, we present experiments on the classical two-dimensional Fisher–KPP equation, a widely studied reaction-diffusion model. The equation is given by

$$u_t = \Delta u + u(1-u) + S(x,y,t),$$

where $(x,y) \in [-1,1]^2$ and $t \in [0,1]$. The exact solution is selected as $u(x,y,t) = e^{-(x^2+y^2+t)}$, from which the source term $S(x,y,t)$ as well as the initial and boundary conditions can be directly determined.

**Experiment 1: NTK Failure in the random feature model for Nonlinear PDEs.**

In this experiment, we use a two-layer neural network,

$$u_\theta = \frac{1}{\sqrt{1000}} \sum_{k=1}^{1000} a_k \tanh\left(w_k^\top \mathbf{x} + b_k\right),$$

where $\mathbf{x} = (x,y,t)^\top$. The inner parameters $(w_k, b_k)$ are initialized as standard Gaussian random variables and then fixed, and the outer coefficients $a_k$ are initialized uniformly in $(-1,1)$ and optimized by the IGD algorithm. The dataset consists of 50 boundary and 200 interior points (full batch). Training is performed over 100 outer steps of learning rate 0.5 (each with 10 L-BFGS inner steps of learning rate 0.1). We report the relative Frobenius norm of the NTK matrices during training in Figure 5, illustrating that the NTK theory breaks down for the nonlinear (interior) component, as evidenced by significant changes in the NTK matrix throughout training.

**Experiment 2: Convergence in the random feature model.**

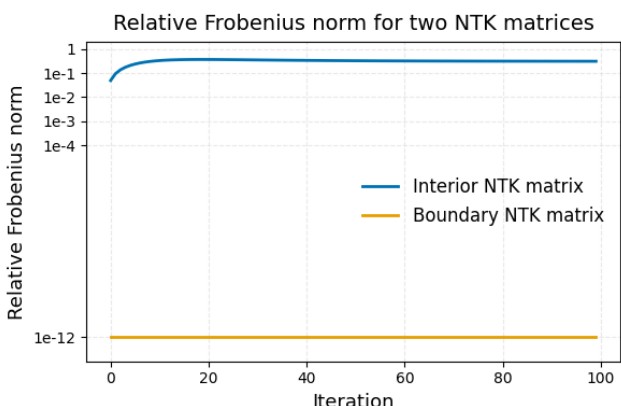

Figure 5: Relative Frobenius norm of two NTK matrices for Fisher–KPP equation.

Here we retain the random feature structure but remove the $\frac{1}{\sqrt{1000}}$ normalization, using $u_\theta = \sum_{k=1}^{1000} a_k \tanh(w_k^\top \mathbf{x} + b_k)$. As before, $(w_k, b_k)$ are fixed after Gaussian initialization and $a_k$ are initialized uniformly in $(-1, 1)$ and trained by IGD. The data comprises 500 boundary points and 10,000 interior points (full batch).

Training is performed for a total of 100,000 steps (2500 IGD outer steps, each with 40 L-BFGS inner steps; outer and inner step sizes are 0.5 and 0.1, respectively). At the end of training, the $\ell_2$-norm of the loss gradient with respect to the parameters is $6.74 \times 10^{-4}$, indicating that $a$ has converged to a critical point (gradient nearly zero), in accord with the theoretical results presented in Theorem 3.

**Experiment 3: IGD demonstrates superior stability to Adam under large step sizes**

We compare the performance of IGD and Adam in solving the Fisher–KPP equation using a four-layer fully connected neural network with 100 neurons per hidden layer. The biases are initialized to zero, and all other trainable parameters are initialized using Xavier normal initialization. The training dataset consists of 500 boundary points and 20,000 interior points, with batch sizes of 32 and 256 for the boundary and interior, respectively.

Training is performed for 200,000 steps. Specifically, IGD is run for 5,000 outer iterations with a learning rate of 0.1, each comprising 40 inner L-BFGS steps (also with learning rate 0.1). Adam is trained for the full 200,000 steps with a fixed learning rate of 0.1. As shown in Figure 6, Adam's loss curve exhibits substantial oscillations during training, whereas the loss for IGD decreases smoothly and steadily, highlighting the superior stability of IGD. In addition, Figure 7 presents the solutions obtained by IGD alongside the exact solutions at three representative time points. The results demonstrate that IGD yields solutions in close agreement with the exact solution.

### I.4 EXPERIMENT TO VALIDATE THEOREM 4

To verify the convergence result stated in Theorem 4, we solve the following PDE within the PINN framework:

$$-\mathrm{div}\left((1 + u^2)\nabla u\right) + q(x)u + h(u) = f(x), \quad q \geq 0, \quad h(u)u \geq 0, \quad (x, y) \in B(0, 1),$$

with homogeneous Dirichlet boundary conditions.

For this experiment, we set $q = 1$ and $h(u) = u^3$, and the chosen exact solution is

$$u(x, y) = 1 - x^2 - y^2.$$

We can canculate the corresponding source term $f(x, y)$ accordingly. The neural network employed is a random feature model, specifically a two-layer network given by

$$u_\theta = \sum_{k=1}^{100} a_k \tanh\left(w_k^\top \mathbf{x}\right),$$

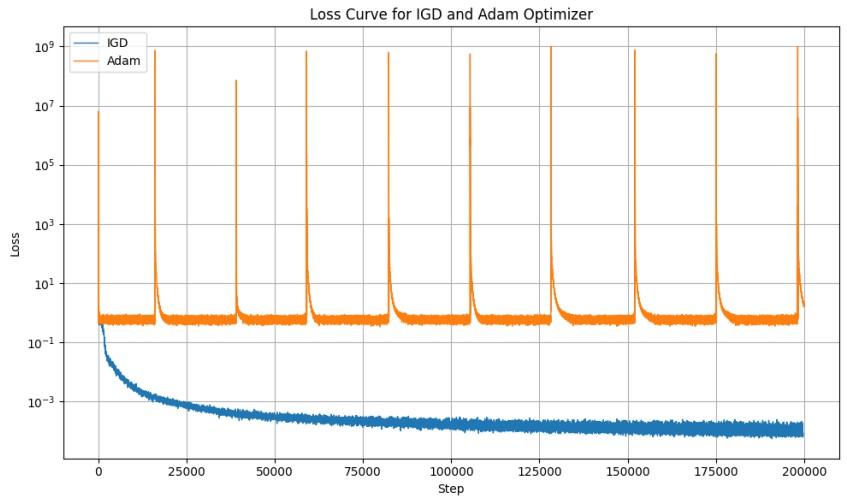

Figure 6: Loss curves for IGD and Adam on Fisher–KPP equation.

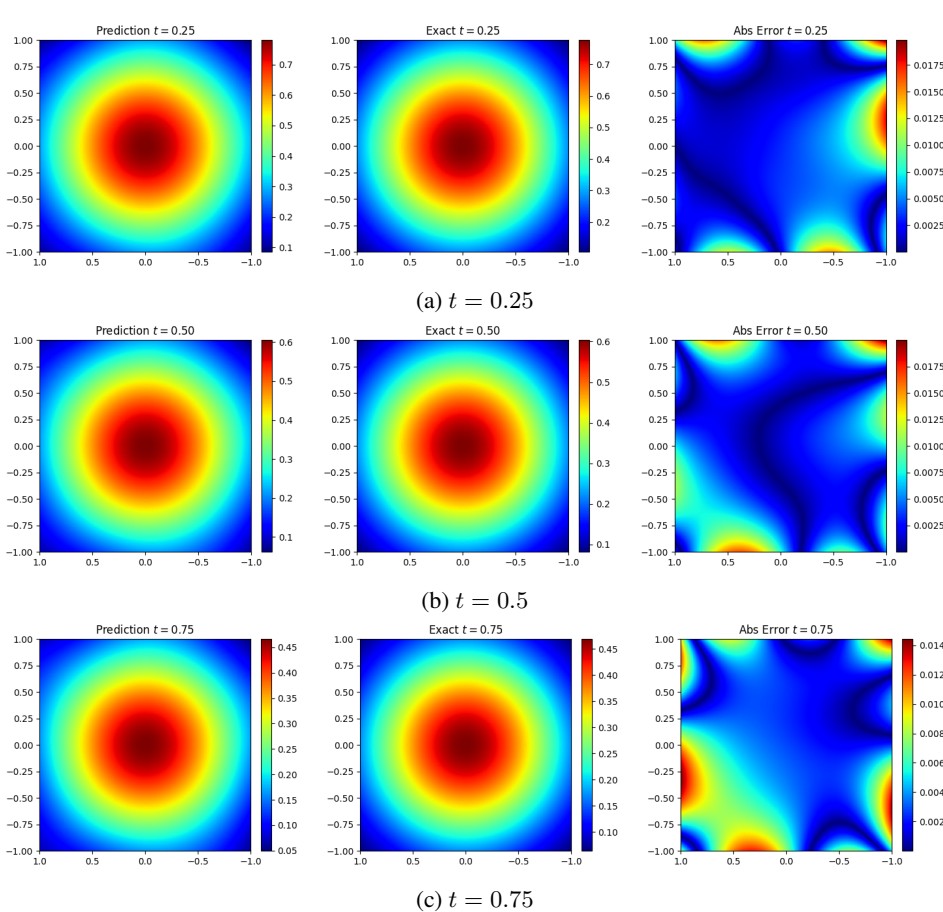

Figure 7: Numerical solutions obtained by IGD at 3 different time points for Fisher–KPP equation.

where $\mathbf{x} = (x, y)^{\mathsf{T}}$, $a_k$ denotes the trainable outer parameters. Only the residual loss is considered, with training points sampled uniformly from $10,000$ locations inside $B(0, 1)$. The homogeneous Dirichlet boundary condition can be enforced by multiplying the network output by a cutoff function $\varphi$ that vanishes on the boundary.

We optimize the outer parameters $a$ using the IGD algorithm sufficiently to ensure convergence. Specifically, IGD updates are performed with a step size of $0.5$ for outer steps, and at each step the subproblem is solved by L-BFGS with a step size of $0.1$ for $40$ inner iterations. Under this protocol, we test the convergence behavior starting from different random initializations of parameters. We report the Euclidean 2-norm of the residual loss gradient with respect to $a$ at the end of training for each initialization, as shown in Table 4. The results confirm the convergence predicted by Theorem 4.

Table 4: Loss gradient norms and IGD steps for different initializations.

| Initialization | $\|\nabla \mathcal{J}(a)\|_2$ | Outer IGD Steps |
|---|---|---|
| Xavier normal for $w_k$, Xavier uniform for $a_k$ | $2.44 \times 10^{-4}$ | 10000 |
| Standard normal for $w_k$, uniform $[-1, 1]$ for $a_k$ | $1.45 \times 10^{-4}$ | 200000 |
| LeCun normal initialization for both $w_k$ and $a_k$ | $1.37 \times 10^{-4}$ | 200000 |

We note that the latter two initializations require more IGD steps to reach convergence. This is because their initial losses are relatively large, resulting in longer optimization trajectories.

## J   Convergence Analysis for SGD

In this section, we investigate whether the convergence results established for full-batch optimization in the main text (Theorem 1 and Theorem 3) can be extended to stochastic gradient descent (SGD). We specifically compare convergence behaviors under SGD for linear and nonlinear PDEs, and highlight the key challenges and open questions arising in the stochastic setting.

### J.1   SGD Convergence for solving Linear PDEs

As discussed throughout this work, convergence analysis for linear PDEs is fundamentally based on NTK theory. Notably, Xu & Zhu (2024) demonstrated that, in supervised learning, one-pass SGD with streaming data—where each iteration samples a fresh, non-repeating point from a continuously distributed dataset—admits a deterministic limit kernel. This insight strongly motivates the use of NTK-based arguments for analyzing the convergence of overparameterized neural networks for linear PDEs within the PINN framework.

However, extending rigorous theoretical results to PINNs in the SGD setting is significantly more complex. Formal proofs demand careful treatment of the interplay between the data distribution, sampling procedure, and network overparameterization, which is beyond the scope of this work. We leave such comprehensive theoretical analysis for future studies.

### J.2   SGD Convergence for Nonlinear PDEs

For nonlinear PDEs, NTK theory is generally inapplicable, and our main text relies instead on the Łojasiewicz inequality for convergence analysis. Recently, works such as Dereich & Kassing (2021); An & Lu (2023) have established convergence guarantees for SGD under certain conditions, notably the boundedness of trajectories, by leveraging the Łojasiewicz inequality as the key tool. While these results provide a natural foundation, applying them directly in the PINN context presents new challenges. In standard supervised learning, the required trajectory conditions can hold with probability one under suitable assumptions; however, for PINNs, it is only straightforward to establish that the probability of bounded SGD trajectories is positive, without control over its magnitude. Quantifying this probability and fully characterizing convergence probabilities remains an open question in the PINN setting.

In summary, while the foundational tools for extending convergence results to SGD exist, a complete understanding—especially for nonlinear PDEs—requires further investigation. In particular, characterizing the likelihood of favorable SGD behavior in the PINN setting represents an important direction for future research.

