# OpenReview forum: "Convergence Guarantees for Gradient-Based Training of Neural PDE Solvers: From Linear to Nonlinear PDEs"
_ICLR.cc/2026/Conference — Submitted to ICLR 2026_

### Official Review · Reviewer_zDM4 · 2025-10-30

**Soundness:** 3
**Presentation:** 3
**Contribution:** 3
**Rating:** 6
**Confidence:** 3

**Summary:**

This paper establishes convergence guarantees for neural network-based PDE solvers, specifically PINNs and the Deep Ritz method. For linear PDEs, the authors extend the NTK framework beyond second-order cases to general admissible linear operators. For nonlinear PDEs, the authors use the Lojasiewicz inequality under a random feature model to prove convergence to critical points for both gradient flow and implicit gradient descent, without requiring strong overparameterization. The theoretical results are supported by numerical experiments on the Burgers, Allen-Cahn, and Fisher-KPP equations.

**Strengths:**

1. The paper provides a unified treatment covering both linear and nonlinear PDEs, and both PINNs and Deep Ritz methods under a single theoretical framework.
2. The proofs are detailed, and the use of Lojasiewicz inequality for nonlinear PDEs is well-motivated. Lemma 1's linear independence result for tanh based networks is a useful technical contribution
3. The numerical experiments on Burgers', Allen-Cahn, and Fisher-KPP equations demonstrate that the theory captures relevant aspects of optimization behavior.
4. Theorem 3 and 4's identification of implicit regularization in random feature models is a valuable observation that could inform practical algorithm design

**Weaknesses:**

1. The scope is a bit limited for random feature model. The main results for nonlinear PDEs (Theorems 3, 4) only apply when inner-layer parameters are frozen. This is a severe restriction since most practical neural PDE solvers train all parameters. The paper acknowledges this in Section H.2 but doesn't provide a clear path forward. How restrictive is this assumption in practice? Can the authors provide empirical evidence comparing random feature models to fully-trained networks on the test problems?
2.  The experiments use implicit gradient descent with specific hyperparameters, but the theory covers gradient flow and generic implicit gradient descent. What convergence rate does Proposition 1 predict for the specific experimental settings? Do the observed convergence rates match theoretical predictions?
3. The convergence results assume the neural network can represent the solution well, but this isn't rigorously analyzed. For nonlinear PDEs with complex solutions (say Allen-Cahn with sharp interfaces), is the two-layer tanh network architecture sufficient?
4. Assumption 1 is needed for Theorem 3. While the paper claims this is "naturally satisfied for evolutionary PDEs," it's unclear how restrictive this is for general domains. What happens for domains without flat boundary portions?

**Questions:**

1. In Theorem 1, how tight is the overparameterization requirement? The bound grows rapidly with dimension and operator order. Are there known lower bounds showing this is necessary?
2. For Theorem 3, what is the typical Lojasiewicz exponent for the loss functions in your experiments? How does this affect the convergence rate in practice?
3. The paper proves convergence under "almost sure" initialization conditions. How robust are the results to initialization in practice? What happens with common initialization schemes like Xavier or He initialization?
4. Section 4.3 considers specific nonlinear operators in Eq 11. Can the interior coercivity analysis be extended to other important nonlinear PDEs, such as Navier-Stokes equations?
5. Can the convergence analysis framework be extended to the Neural Galerkin method or TENG (https://proceedings.mlr.press/v235/chen24ad.html)? Both use sequential-in-time optimization with projections onto neural network manifolds. Would the Lojasiewicz approach apply, and what additional assumptions would be needed?
6. How does the implicit regularization effect in random feature models manifest in the experiments?

---

> ### Author Response · Authors · 2025-11-19
>
> We sincerely thank the reviewer for their thoughtful and encouraging feedback. We greatly appreciate your recognition of the unified theoretical framework, the detailed proofs, and the technical contributions such as the linear independence result in Lemma 1. Your positive comments on our use of the Łojasiewicz inequality for nonlinear PDEs, as well as the practical significance of Theorems 3 and 4, are very motivating for us.  Thank you again for your careful reading and supportive assessment.
>
>
> >### Point 1:
> >
> > The scope is a bit limited for random feature model. The main results for nonlinear PDEs (Theorems 3, 4) only apply when inner-layer parameters are frozen. This is a severe restriction since most practical neural PDE solvers train all parameters. The paper acknowledges this in Section H.2 but doesn't provide a clear path forward. How restrictive is this assumption in practice? Can the authors provide empirical evidence comparing random feature models to fully-trained networks on the test problems?
>
> ### Reply:
>
> Thank you for your insightful comment. You are indeed correct that focusing on the random feature model, where inner-layer parameters are frozen, is a significant limitation compared with the settings of most practical neural PDE solvers where all parameters are trained. As discussed in Section H.2, for fully trainable two-layer networks, the Łojasiewicz inequality implies that either the parameters converge to a critical point or their norm diverges to infinity. Without further assumptions, we cannot exclude the latter case. However, in practice, techniques such as gradient clipping and implicit regularization can help prevent parameter explosion, allowing convergence to be observed in real training scenarios. We will continue to investigate and develop a more precise theoretical understanding in this direction.
>
> Regarding empirical evidence, we also observe that for nonlinear PDEs, the random feature model performs significantly worse than fully trained networks. As shown in our experiments, the loss curves for the random feature model decrease much more slowly, and the accuracy is limited compared to the fully trainable case. We acknowledge this limitation—this paper is just an initial step, and we plan to conduct more comprehensive studies on convergence behavior under more practical and expressive network architectures in future work. Thank you again for your valuable feedback.
>
>
> >### Point 2:
> >
> > The convergence results assume the neural network can represent the solution well, but this isn't rigorously analyzed. For nonlinear PDEs with complex solutions (say Allen-Cahn with sharp interfaces), is the two-layer tanh network architecture sufficient?
>
> ### Reply:
>
> Thank you for your thoughtful comment. You are correct that our convergence results for nonlinear PDEs (as in Theorem 3) only guarantee convergence to a critical point, and unlike the linear case (Theorem 1), they do not ensure that the loss will converge to zero. This indeed means that, even after sufficient training, the critical point achieved may correspond to a non-negligible loss, particularly if the network architecture cannot accurately represent the solution. Therefore, for nonlinear PDEs with complex features, such as sharp interfaces in the Allen--Cahn equation, a simple two-layer tanh network may not always have sufficient expressive power to approximate the true solution well. This is an important aspect and a limitation of our current analysis. Thank you again for highlighting this point.
>
> >### Point 3:
> >
> > Assumption 1 is needed for Theorem 3. While the paper claims this is "naturally satisfied for evolutionary PDEs," it's unclear how restrictive this is for general domains. What happens for domains without flat boundary portions?
>
> ### Reply:
>
> Thank you for pointing out this important aspect. Our assumption requires that the domain possesses at least a small segment of flat boundary, which is actually less restrictive than assuming local manifold properties and thus covers a fairly broad class of domains. For domains that do not satisfy this assumption, and where interior coercivity for the nonlinear PDE operator cannot be established using tools such as the Sobolev inequality (as discussed in Section 4.3), we currently do not have any convergence guarantees. This remains an open problem, and we plan to investigate broader conditions in future work. Thank you again for highlighting this limitation.

---

> > ### Author Response · Authors · 2025-11-19
> >
> > >### Point 4:
> > >
> > > In Theorem 1, how tight is the overparameterization requirement? The bound grows rapidly with dimension and operator order. Are there known lower bounds showing this is necessary?
> >
> > ### Reply:
> >
> > Thank you for your insightful question. In practice, the overparameterization bound given in Theorem 1 is usually not treated as a strict guideline; instead, practitioners typically ensure the network has "sufficiently many" parameters. To the best of our knowledge, there are currently no established lower bounds describing the minimal necessary level of overparameterization for such problems. We agree this is an interesting and important direction for future research, and we plan to explore it further. Thank you again for raising this point.
> >
> >
> > >### Point 5:
> > >
> > > The paper proves convergence under "almost sure" initialization conditions. How robust are the results to initialization in practice? What happens with common initialization schemes like Xavier or He initialization?
> >
> > ### Reply:
> >
> > Thank you for your question. Our results hold for all commonly used random initialization schemes in practice, including Xavier and He initialization. The key point in the proofs of Theorems 3 and 4 is that the required property of the inner-layer parameters (as stated in Definition 3) holds with probability one under any initialization distribution. Both Xavier and He initializations satisfy this condition. Therefore, our convergence results remain valid under these widely used initialization schemes.
> >
> > >### Point 6:
> > >
> > > Section 4.3 considers specific nonlinear operators in Eq 11. Can the interior coercivity analysis be extended to other important nonlinear PDEs, such as Navier-Stokes equations?
> >
> > ### Reply:
> >
> > Thank you for your question. In Section 4.3, we illustrate interior coercivity with two representative classes of nonlinear PDEs. Whether the interior coercivity analysis can be extended to other nonlinear PDEs, such as the Navier-Stokes equations, depends on the specific structure of the operator and the domain. For some equations like Navier-Stokes, interior coercivity may not hold in general. Therefore, the convergence analysis for PINN and related solvers on such problems—especially over domains that do not satisfy Assumption 1—would require dedicated study. We appreciate your suggestion and agree this is an important direction for further research.
> >
> >
> > Thank you very much for your thoughtful and professional questions. Your feedback is both valuable and encouraging for our work. While we recognize that our current theoretical framework has its limitations, we see this as an important starting point. We remain committed to advancing our research and developing new theoretical tools, with the sincere hope that future work will more fully address the important issues you have highlighted.

---

### Official Review · Reviewer_cb9M · 2025-10-31

**Soundness:** 2
**Presentation:** 2
**Contribution:** 2
**Rating:** 2
**Confidence:** 4

**Summary:**

This paper presents convergence guarantees for 2 layer PINNs for approximating linear and non-linear PDEs using gradient-descent based methods, where for nonlinear PDEs, the hidden layer parameters are initialized from fixed distributions and left untrained. Experiments are conducted with the Burgers' equation to support theoretical claims.

**Strengths:**

(1) The manuscript is well written with a good motivation on the missing convergence properties for non-linear PDEs for PINNs. Especially the PDE setting considered in the paper is concise and on point.
(2) The manuscript touches on the concrete missing points in the convergence theory and error bounds for PINNs / deep Ritz.

**Weaknesses:**

(1) The paper heavily relies on the random-feature weight initialization, but does not provide a through literature review on the random features one can use.
 - Fourier random feature approaches should be cited for data-agnostic cases, e.g.
(1.1) Li, Zhu, Jean-Francois Ton, Dino Oglic, and Dino Sejdinovic. 2021. “Towards a Unified Analysis of Random Fourier Features.” Journal of Machine Learning Research 22 (108): 1–51.
 - Data-driven approaches for random features are also missing; e.g.
(1.2) Bolager, Erik L, Iryna Burak, Chinmay Datar, Qing Sun, and Felix Dietrich. 2023. “Sampling Weights of Deep Neural Networks.” Advances in Neural Information Processing Systems 36: 63075–116.

(2) Only last layer parameters are trained using iterative gradient-descent methods, which turns the networks into linear models (e.g., like finite element method approaches in classical scientific computing). This class is provably separated form general neural networks, and thus the results in the paper are more about random feature methods (with problem-agnostic weights), rather than PINNs trained wtih gradient descent or random features with data-driven weights. This is not discussed, and should be cited, e.g.:
(2.1) Wu, Lei, and Jihao Long. 2022. A Spectral-Based Analysis of the Separation between Two-Layer Neural Networks and Linear Methods. Journal of Machine Learning Research, vol. 23 (1).

For the claim of "establishing a systematic convergence theory for neural PDE solvers across both linear and nonlinear regimes", the random feature model is far too restrictive.

(3) The claim: 'using random-features as regularization' is not supported with additional experiments. For example, data-driven random features may be effective to sample more basis-functions in the shock regions (see "Datar, Chinmay, Taniya Kapoor, Abhishek Chandra, et al. 2024. “Solving Partial Differential Equations with Sampled Neural Networks.” arXiv:2405.20836. Preprint, arXiv, May 31, 2025.")

(4) The operator $\mathcal{L}$ is introduced as "a differential operator that may be linear or nonlinear" in line 119, but then redefined as linear operator only in line 170.

(5) l 2025 "similar techniques apply to a broader class of linear operators. Further details are omitted for brevity." This is a claim that should not just "omitted for brevity", especially if there is an extensive appendix to the paper already. Either the claim is supported by additional results (in the appendix, to not make the main paper longer) or the claim must be removed. Brevity alone is not enough to omit results; this also holds for Remark 5. The paper is already 37 pages long, it is not clear why results were simply omitted. If they can be found in the appendix, this must be stated in the main text.

(6) Some of the proofs are not precise enough. Example: l.422, "Fix any choice of inner-layer parameters" while the proof then requires specific choices of inner-layer parameters (l427). Similarly, for l328, "for randomly initialized inner parameters" does not specify the distribution. The longer versions of the proofs in the appendix should be referenced each time, to clarify that these are not the entire proofs.

**Questions:**

(1) When does the random feature model become restrictive during training (of the last layer), or for which problems?
(2) Could you elaborate on the random-features acting as regularization and explain the limitations of this? It is a very uncommon type of regularization. Usually, the training algorithm (Adam, SGD) itself acts as the "implicit regularization", while training the inner weights.
(3) Do the convergence bounds also hold with a relaxation when optimizing the hidden layer parameters as well for PINNs with the 2-layer architecture?

---

> ### Author Response · Authors · 2025-11-19
>
> Thank you very much for your detailed reading and for highlighting the strengths of our work. We sincerely appreciate your positive comments on both the motivation and the clarity of our manuscript.
>
> >### Point 1:
> >
> > The paper heavily relies on the random-feature weight initialization, but does not provide a thorough literature review on the random features one can use.
>
> ### Reply:
>
> Thank you for highlighting this important point. We agree that providing a more thorough literature review on random features would enhance the completeness of our work. We will address this in a future revision by including relevant references and discussions to give readers a broader view of possible random feature choices. We appreciate your constructive suggestion.
>
> >### Point 2:
> >
> > The operator is introduced as "a differential operator that may be linear or nonlinear" in line 119, but then redefined as a linear operator only in line 170.
>
> ### Reply:
>
> Thank you for your careful reading and for pointing out this potential confusion. Our intention was to first introduce the general differential operator $\mathcal{L}$ on line 119, allowing for both linear and nonlinear cases. Later, on line 170, we specify a subclass of linear differential operators for which Theorem 1 and the NTK-based convergence analysis apply.
>
> In Section 4, including Theorem 3, the convergence results do not require $\mathcal{L}$ to be linear—the operator can be either linear or nonlinear in that context. Thus, the introduction on line 119 sets the general stage, while the definition on line 170 is for clarity regarding the precise assumptions under which certain results (like Theorem 1) hold. These two usages do not conflict, and we will clarify this distinction in future revisions to avoid confusion. Thank you again for your helpful feedback.
>
> >### Point 3:
> >
> > l 2025 "similar techniques apply to a broader class of linear operators. Further details are omitted for brevity." This is a claim that should not just "omitted for brevity", especially if there is an extensive appendix to the paper already. Either the claim is supported by additional results (in the appendix, to not make the main paper longer) or the claim must be removed. Brevity alone is not enough to omit results; this also holds for Remark 5. The paper is already 37 pages long, it is not clear why results were simply omitted. If they can be found in the appendix, this must be stated in the main text.
>
> ### Reply :
>
> Thank you for your detailed feedback. We understand your concern about omitting details, especially given the length of the manuscript. Our intention in stating that “similar techniques apply to a broader class of linear operators” was to provide guidance to interested readers that the proof strategy is adaptable, without overwhelming the main text with highly technical variants. Theorem 1 already covers a unified and substantial class of differential operators, as specified in Definition 1, and the proof is constructed to be as general as possible while maintaining clarity and readability.
>
> We agree that it is not always necessary or practical for a theorem to cover every possible case within its statement, especially when extensions follow similar lines. Thank you for raising this point and helping us improve the clarity and completeness of our presentation.
>
> >### Point 4:
> >
> > Some of the proofs are not precise enough. Example: l.422, "Fix any choice of inner-layer parameters" while the proof then requires specific choices of inner-layer parameters (l.427). Similarly, for l.328, "for randomly initialized inner parameters" does not specify the distribution. The longer versions of the proofs in the appendix should be referenced each time, to clarify that these are not the entire proofs.
>
> ### Reply:
>
> Thank you for your detailed comments on the precision of the proofs. We would like to clarify two points:
>
> First, regarding the distribution of the randomly initialized inner parameters, lines 373–374 of the main text specify the random initialization scheme we use. Thus, the distribution is clearly stated in the manuscript.
>
> Second, our proof approach is as follows: every choice of inner-layer parameters satisfies the formula in line 423, but line 428 refers to the property defined in Definition 3. The parameters that satisfy both are exactly those sampled according to our random initialization, which, as shown, occurs with probability one under the stated distribution.

---

> > ### Author Response · Authors · 2025-11-19
> >
> > > ### Point 5:
> > >
> > > When does the random feature model become restrictive during training (of the last layer), or for which problems?
> > >
> >
> > ### Reply:
> >
> > Thank you for the insightful question. We acknowledge that, in practice, the random feature setting is rarely used for solving nonlinear PDEs, as it can be restrictive regarding representational capability. In our study, we adopted this framework primarily to establish our convergence Theorem 3—this is, to the best of our knowledge, the first theoretical result on convergence for nonlinear PDEs in the PINN literature. Although this setup is more limited compared to practical architectures where all parameters are trainable, it provides a theoretically manageable starting point for understanding convergence in this challenging setting. Going forward, we aim to extend our results to broader and more practical scenarios.
> >
> > > ### Point 6:
> > >
> > > Could you elaborate on the random-features acting as regularization and explain the limitations of this? It is a very uncommon type of regularization. Usually, the training algorithm (Adam, SGD) itself acts as the "implicit regularization", while training the inner weights.
> > >
> >
> > ### Reply:
> >
> > Thank you for your thoughtful question. In our context, the “regularization” effect refers to the architectural constraint of fixing the inner-layer parameters, which limits the function space and governs the optimization process. Specifically, when training only the outer coefficients in the random feature model, their values remain in a bounded region during gradient descent, preventing gradient explosion and stabilizing the optimization dynamics. This is a milder and less flexible form of regularization compared to implicit regularization from algorithms like SGD or Adam, as it stems from the model structure itself rather than the training procedure. However, it is sufficient to ensure control of the optimization in our theoretical setting. We recognize that this type of regularization has its limitations in terms of model expressivity, and it represents one trade-off in our analysis for establishing rigorous convergence theory.
> >
> >
> > > ### Point 7:
> > >
> > > Do the convergence bounds also hold with a relaxation when optimizing the hidden layer parameters as well for PINNs with the 2-layer architecture?
> > >
> >
> > ### Reply:
> >
> > Thank you for raising this important point. When considering fully trainable two-layer networks (with optimizable inner-layer parameters), the situation becomes much more complex. In general, as shown by the Lojasiewicz inequality, we can only conclude that the parameters either converge to a critical point or their norms diverge to infinity. Without additional constraints or regularization, we cannot guarantee convergence in this broader setting. However, if practical techniques such as gradient clipping or implicit regularization are used to prevent parameter explosion, convergence can also be achieved. Developing a more comprehensive theoretical framework to address these more realistic, fully-trainable scenarios is part of our ongoing and future research agenda.

---

### Official Review · Reviewer_Ez4C · 2025-10-31

**Soundness:** 3
**Presentation:** 3
**Contribution:** 2
**Rating:** 2
**Confidence:** 3

**Summary:**

The paper provides a systematic analytical framework to establish convergence rates and guarantees, bridging the gap between existing NTK-based results for linear problems and new, though restricted, theoretical results for the challenging non-convex landscapes of nonlinear PDE solvers.

**Strengths:**

1. The novel application of the Łojasiewicz inequality to analyze the convergence of the non-convex loss functions encountered when solving nonlinear PDEs with neural networks.
2. The article provides a  technically solid extension of the NTK theory for linear PDEs.
3. This theoretical article presents mathematically rigorous convergence results that span a wide class of PDEs.

**Weaknesses:**

1. The core weakness is proving convergence for nonlinear problems only under the RFM, where only the last layer weights are trained.
2. The empirical validation of the derived theory is very limited, focusing only on a single 1-D nonlinear PDE and lacking higher-dimensional or real-world benchmarks.
3. For nonlinear PDEs, the proven convergence is only to a critical point of the loss function. A critical point is merely a state where the gradient is zero, which could be a local minimum.
4. The entire analysis depends on the properties of a two-layer network with a $\tanh$ activation function. While this is common in theoretical work, practical high-performing solvers often use deeper architectures and $\mathrm{ReLU}$ or $\mathrm{SiLU}$ activations.

**Questions:**

See the weaknesses.

---

> ### Author Response · Authors · 2025-11-19
>
> Thank you very much for your thorough review. We greatly appreciate that you took the time to carefully read our work and provide an accurate summary of our main contributions. We are especially grateful for your recognition of our use of the Łojasiewicz inequality in analyzing convergence for non-convex loss functions, as well as your appreciation for our technical extension of NTK theory and rigorous convergence results for a broad class of PDEs.
>
>
> >### Point 1:
> >
> > The core weakness is proving convergence for nonlinear problems only under the RFM, where only the last layer weights are trained.
>
> ### Reply:
>
> Thank you for your thoughtful comment. We agree with your observation and acknowledge this limitation. Our convergence guarantees for nonlinear problems are currently established under the random feature model, where only the last layer is trained. This work serves as an initial step—previous results mainly focused on NTK-based analysis for linear PDEs.
>
> For fully trainable two-layer networks, the Łojasiewicz inequality suggests either convergence to a critical point or divergence of the parameter norm. Without additional assumptions, it is theoretically difficult to exclude the latter. Nevertheless, in practical training, techniques like gradient clipping or implicit regularization can help prevent parameter explosion and support convergence.
>
> While our results represent only the first step, we believe they lay a foundation for deeper and more comprehensive analysis of nonlinear PDE solvers, and we are committed to further advancing this line of research in the future. Thank you for helping us clarify the scope and implications of our work.
>
> >### Point 2:
> >
> > The empirical validation of the derived theory is very limited, focusing only on a single 1-D nonlinear PDE and lacking higher-dimensional or real-world benchmarks.
>
> ### Reply:
>
> Thank you for your comment. As mentioned in line 449 of the main text, we have also conducted experiments on two-dimensional PDEs, specifically the Allen–Cahn equation and the Fisher–KPP equation. The details of these experiments and their results can be found in the appendix.
>
> >### Point 3:
> >
> > For nonlinear PDEs, the proven convergence is only to a critical point of the loss function. A critical point is merely a state where the gradient is zero, which could be a local minimum.
>
> ### Reply:
>
> Thank you for highlighting this important point. We agree that our current results show convergence only to a critical point, which could indeed be a local minimum or another stationary point. Our results indicate that if, during the training process for nonlinear PDEs, the method enters a neighborhood of a local minimum, it will converge. This work represents an initial step for nonlinear PDE solvers—previously, even convergence to a critical point was not theoretically guaranteed. In future work, we plan to investigate saddle point dynamics and explore sufficient conditions for ensuring convergence specifically to local minima, further strengthening the theory. We appreciate your suggestion.
>
> >### Point 4:
> >
> > The entire analysis depends on the properties of a two-layer network with a specific activation function. While this is common in theoretical work, practical high-performing solvers often use deeper architectures and ReLU or SiLU activations.
>
> ### Reply:
>
> Thank you for your constructive feedback. As discussed in both the main text and the appendix, we have extended our analysis to deeper networks and provided corresponding convergence results for multi-layer architectures.
>
> Regarding activation functions, our results are not limited to $\tanh$ but apply to a broad class of analytic activation functions, as also mentioned in the paper (Remark 1). For activations such as ReLU and SiLU, convergence for solving a wide class of linear PDEs by PINNs using (implicit) gradient descent can often be established via NTK theory. However, for nonlinear PDEs, our convergence results rely on the Łojasiewicz inequality, which requires the activation function to be analytic, so it does not directly apply to networks using ReLU.
>
> Overall, we have aimed to provide results as general as possible within the current mathematical framework. While it is difficult for a mathematical theorem to cover all practical scenarios, we hope our work offers valuable theoretical insight and a basis for future research. Thank you again for your valuable suggestions.
>
> Thank you again for your careful review and thoughtful feedback. We sincerely appreciate your constructive suggestions. We look forward to continuously improving this line of research and hope our results can provide useful insights for both theory and practice in neural PDE solvers.

---

> > ### Comment · Reviewer_o3z7 · 2025-11-26
> >
> > The reviewer appreciates the authors’ response.Since the paper relies on Lemma G.6 from [Du et al. (2019a)](https://proceedings.mlr.press/v97/du19c.html), I would encourage the authors to explicitly state the required assumptions on the training data. In [Du et al. (2019a)](https://proceedings.mlr.press/v97/du19c.html), these conditions (e.g., non-parallel data leading to a strictly positive smallest eigenvalue of the NTK under sufficient nonlinear activation) are nowadays standard and easy to satisfy, but they are necessary for the lemma to hold. If the proof indeed depends on this lemma, the corresponding data assumptions should be clearly specified.
> >
> > Regarding the technical contribution: while the paper is correct as a theoretical analysis of shallow networks in the NTK regime, the results largely follow the standard NTK framework. Since NTK theory does not capture feature learning, its ability to reveal new insights is inherently limited. Consequently, the current analysis, although valid, does not appear to yield conceptual advances beyond existing NTK results, nor does it provide additional practical or theoretical implications.
> >
> > Therefore, my evaluation remains unchanged.

---

### Official Review · Reviewer_o3z7 · 2025-11-03

**Soundness:** 2
**Presentation:** 2
**Contribution:** 1
**Rating:** 2
**Confidence:** 3

**Summary:**

This paper studies the training dynamics and convergence of neural PDE solvers using NTK and Łojasiewicz inequality under the random-feature model for linear and nonlinear PDEs, respectively.

**Strengths:**

- The paper is overall well written and easy to follow, though a few key notations could be introduced earlier for clarity.
- The work approaches the study of training dynamics and convergence from a rigorous theoretical perspective.

**Weaknesses:**

- Convergence of two-layer MLPs under the NTK regime is already well established. Theorem 1 merely applies this known result under an additional regularization on the differential operator. Moreover, the theorem could fail if the smallest eigenvalues are zero, and the authors does not discuss this possibility and even not officially define the Gram matrices before or within the theorem statement until next subsection.

- Sections 4.1–4.3 train only the second layer while freezing the first-layer weights, making the model linear in parameters. The resulting analysis essentially reduces to somewhat least-squares convergence under mild regularity (coercivity or PL-type) conditions, offering only incremental extensions of classical optimization theory rather than new insights into nonlinear or deep neural PDE solvers.

- The paper provides no experiments or numerical evidence verifying whether the theoretical assumptions hold in practice, leaving the practical relevance of the results unclear.

- The residual $r_\theta$ and its associated NTK-based Gram matrix, both central to Theorem 1, are not formally defined until Section 3.2.2, and only through an illustrative example. This late introduction makes the theoretical study difficult to follow.

**Questions:**

- In Theorem 1, you assume positive smallest eigenvalues $\lambda_0, \tilde{\lambda}_0 > 0$. How realistic is this assumption in practice, and can you provide numerical evidence or bounds showing these eigenvalues are indeed positive for typical PINN setups?

- Since the paper focuses on implicit gradient descent, have you explored whether the same convergence behavior holds for explicit gradient descent, which is what practitioners actually use?

---

> ### Author Response · Authors · 2025-11-19
>
> Thank you very much for your time and thoughtful feedback on our paper. We appreciate your recognition of our theoretical approach and your positive comments on the clarity of the writing. Your remarks regarding notation and presentation are helpful, and we also value your honest assessment of the paper’s contributions. We address your comments carefully below.
>
> >### Point 1:
> >
> > Convergence of two-layer MLPs under the NTK regime is already well established....
> ### Reply:
>
> First, we would like to clarify that the condition “the smallest eigenvalues are positive” is not assumed in our theorem, but rather it is a result that we prove. Specifically, in the full proof of Theorem 1 provided in the appendix, the very first step is to show that the minimum eigenvalue of the Gram matrix at initialization is greater than zero with high probability (Lemma 3&4 in the appendix). This positivity is thus not a precondition, but a conclusion derived from our analysis.
>
> Second, regarding the presentation of the NTK matrix: We understand your concern about the timing of introducing the explicit form of the NTK. Due to space constraints, we opted to provide the precise definition and full expression for the NTK matrix in the appendix. Our intention was to maintain readability in the main text. Readers interested in the details can find the complete form and related proofs in the appendix, while those primarily interested in the convergence result of Theorem 1 can proceed without additional technical complexity.
>
> Lastly, we would like to clarify the novelty of Theorem 1. While it is true that previous convergence results have been established for two-layer MLPs under the NTK regime, these results have been limited to second-order elliptic PDEs. Our main contribution is a nontrivial extension: we generalize the proof to hold for PDEs involving a much broader family of linear differential operators. This required us to generalize every step of the original analysis and rigorously formalize these generalizations. We believe this is a meaningful advancement, as it opens the door for the application of NTK-based analysis to a significantly wider class of problems.
>
> We appreciate the opportunity to clarify these points and thank you again for your helpful suggestions.
>
> >### Point 2:
> >
> > Sections 4.1–4.3 train only the second layer while freezing the first-layer weights, making the model linear in parameters....
>
> ### Reply:
>
> Thank you for your thoughtful observations on our analysis approach in Sections 4.1–4.3. As you noted, under the random feature model, the neural network becomes linear in its parameters. However, the PDEs considered in Section 4 are nonlinear, resulting in loss functions for solvers such as PINN or Deep Ritz that are highly nonlinear and nonconvex with respect to the parameters—even under the random feature model. Therefore, classical least-squares convergence theory does not directly apply to this setting.
>
> Moreover, our convergence results for nonlinear PDEs (Theorems 3 & 4) are, to the best of our knowledge, new. Previous work has been able to establish convergence only for a specific class of linear PDEs. By extending the analysis to include nonlinear PDEs, we believe our results can provide  theoretical guarantees for the training dynamics of neural PDE solvers in a much broader context.
>
> We appreciate your critical perspective, as it has given us the opportunity to clarify the novelty of our contributions. Thank you for considering this explanation.
>
> >### Point 3:
> >
> > The paper provides no experiments or numerical evidence verifying whether the theoretical assumptions hold in practice, leaving the practical relevance of the results unclear.
>
> ### Reply:
>
> Thank you for raising this important concern about the practical relevance of our theoretical assumptions. As mentioned in our reply to Point 1, the positivity of the NTK minimum eigenvalue is not assumed but rather proven to hold with high probability at initialization. Therefore, our analysis does not rely on unverifiable assumptions regarding the NTK.
>
> In our work, the primary explicit assumption is on the geometry of the PDE domain. This assumption is standard and encompasses a wide variety of domains encountered in real-world problems, ensuring that our convergence result (Theorem 3) is broadly applicable to many nonlinear PDEs of practical interest.
>
> We appreciate your suggestion to further clarify the scope of our assumptions. Thank you for your feedback.

---

> ### Author Response · Authors · 2025-11-19
>
> >### Point 4:
> >
> > The residual $r_{\theta}$ and its associated NTK-based Gram matrix, both central to Theorem 1, are not formally defined until Section 3.2.2, and only through an illustrative example. This late introduction makes the theoretical study difficult to follow.
>
> ### Reply:
>
> Thank you for pointing out this aspect of our presentation. Our intention was to keep the main text as concise and readable as possible, especially given space constraints. For readers mainly interested in the convergence result of Theorem 1, familiarity with the details of the NTK and residual definitions is not strictly necessary, allowing the main ideas to be accessed more directly.
>
> For readers seeking a deeper understanding of Theorem 1 and the precise forms of the NTK, we provided clear references to the appendix in the main text. The appendix contains the detailed definitions and full proofs, with direct links for ease of navigation.
>
> We appreciate your suggestion, and in future revisions we will consider introducing these central objects earlier in the main text or providing a brief summary before their detailed discussion, to further enhance clarity for all readers. Thank you for helping us improve the accessibility of our work.
>
> >### Point 5:
> >
> > In Theorem 1, you assume positive smallest eigenvalues. How realistic is this assumption in practice, and can you provide numerical evidence or bounds showing these eigenvalues are indeed positive for typical PINN setups?
>
> ### Reply:
>
> Thank you for raising this point. As discussed in our reply to Point 1, we do not assume the positivity of the smallest eigenvalues as a precondition. Instead, we rigorously prove in the appendix that the Gram matrix at initialization is positive definite with high probability. This result follows established concentration arguments for neural tangent kernels, ensuring the smallest eigenvalue is strictly positive in typical settings.
>
> Therefore, the positivity of the smallest eigenvalues is not only realistic, but can also be justified theoretically for standard PINN setups. We appreciate your attention to this detail and will make this point clearer in our revision.
>
> >### Point 6:
> >
> > Since the paper focuses on implicit gradient descent, have you explored whether the same convergence behavior holds for explicit gradient descent, which is what practitioners actually use?
>
> ### Reply:
>
> Thank you for this insightful question regarding the practical applicability of our results to explicit gradient descent.
>
> For the convergence result in Theorem 1 (which covers a broad class of linear PDEs using NTK), as noted in our paper’s remark, similar convergence guarantees also hold for explicit gradient descent with a sufficiently small step size; specifically, as long as the step size is below a certain threshold, the result remains valid.
>
> For our results concerning nonlinear PDEs (Theorems 3 & 4), if the step size in explicit gradient descent is chosen appropriately—such as ensuring decrease of the loss at each iteration—the same style of convergence proofs can be applied.
>
> In summary, our theoretical guarantees extend to explicit gradient descent under suitable step size conditions. Nevertheless, since gradient flow can be viewed as a continuous analogue to gradient descent, and implicit gradient methods are often more stable for multi-scale PDEs, we chose to focus on these two frameworks in the main text.
>
> We hope our responses have helped to clarify the key points regarding our work. Thank you again for your thoughtful and constructive feedback, which has allowed us to sharpen both the presentation and the scope of our results. We sincerely appreciate your time and effort in reviewing our paper.

---

### Meta-Review · Area_Chair_io9p · 2025-12-30

**Summary:**

The paper received three negative reviews (rejection) and one mildly positive review (marginally above acceptance) before the rebuttal. The major concerns raised from the reviews include:
1) Several reviewers consider the technical novelty of the theoretical results over prior arts in neural tangent kernel (NTK) and classical optimization theory incremental or unclear.
2) The technical analysis focuses on small, two-layer networks and only trains the last-layer parameters. With the inner-layer parameters frozen, the method’s extension to deeper models widely used in real-world applications is unclear.
3) The paper lacks strong numerical experiments, e.g., higher-dimensional problems and real-world benchmarks, to validate the practical applicability of the theoretical results.

**Reviewer Concerns:**

Regarding the first concern, the rebuttal stated that it extended existing results for elliptic PDEs to a broader family of PDEs and claimed that its results for non-linear PDEs are novel. After reading the reviews and the rebuttal, I am inclined to believe the rebuttal has partially resolved this concern. Still, some minor issues persist, e.g., the extension relies on some model and training assumptions.

Regarding the second concern, the rebuttal emphasized its theoretical implications, viewing it as an initial yet foundational step toward future study. The rebuttal discussed the challenges of relaxing hidden-layer parameters and considered this to be future work. Based on the current form of the rebuttal, I think this remains a concern among several reviewers and would result in more discussion.

Regarding the last concern, the rebuttal clarified its assumptions for practical settings. It reiterated its theoretical contribution and acknowledged certain limitations in real-world scenarios. I think it partially addressed the concern, but more discussions on this issue would be needed if the reviewers and the authors could reach a consensus.

**Reviewer Scores:**

I think the borderline positive (6) review would retain its positive opinion. The other three reviewers (2/2/2) expressed major concerns as summarized above. While the rebuttal have made noticable efforts and might affect the reviewers’ opinions positively, some significant issues with the theoretical analysis and its empirical validation would probably persist after discussion.

---

### Decision · Program_Chairs · 2026-01-26

Reject